# ProMiSE: Protein Multi-State Evaluation Benchmark in Biological Contexts

**Bonjae Ku** [* 1]  **Seeun Kim** [* 1]  **Yubeen Kim** [* 1]  **Hahnbeom Park** [2]  **Chaok Seok** [1]

## Abstract

Proteins are inherently dynamic, with biological functions often emerging from transitions between multiple conformational states. While recent breakthroughs have largely addressed the static structure prediction problem, no systematic benchmark exists to demonstrate how well current models capture functionally relevant dynamics. We introduce ProMiSE, the first benchmark that provides both a dataset and an evaluation scheme, based on native biological assemblies and integrating major conformational change mechanisms—intrinsic, ligand-induced, and protein-induced—within a single curated dataset. We conducted a comprehensive evaluation of state-of-the-art structure prediction models, including AlphaFold3 and recent generative approaches. Our findings reveal that current models exhibit a limited ability to sample intrinsic multi-states and are often insensitive to biological context in induced scenarios. Internal representation analysis suggests that training-data exposure can shift predictions toward dominant conformational states over alternative biologically relevant states, primarily at the structure module. In contrast, results from BioEmu indicate that reducing decoding-stage bias can substantially improve multi-state sampling without major changes to upstream pair representations.

## 1. Introduction

Proteins function through transitions between multiple conformational states, making protein structure prediction inherently a multi-state problem (Henzler-Wildman & Kern, 2007; Guo & Zhou, 2016). Despite substantial progress in static structure prediction (Jumper et al., 2021), current models tend to predict a single dominant structure, leaving their ability to sample multiple biologically relevant conformations largely unexplored. In biologically important systems such as fold-switching proteins or ligand-induced conformational changes, models often favor a single dominant structure rather than recovering multiple experimentally characterized states (Porter & Looger, 2018; Kim & Porter, 2021; Chakravarty & Porter, 2022).

It is essential to consider biological context to understand protein conformational changes. Conformational changes arise from distinct mechanisms tied to different biological contexts—intrinsic fluctuations, ligand binding, and protein–protein interactions—which differ in both structural signatures and functional roles (Henzler-Wildman & Kern, 2007; Guo & Zhou, 2016). However, existing evaluations rarely consider these mechanisms within a unified framework and often rely on monomeric-only context rather than biologically relevant complex context, limiting insight into how models behave across different classes of state transitions.

To address this gap, we introduce ProMiSE, a rigorously curated benchmark for systematically evaluating multi-state protein structure prediction under biologically realistic conditions. ProMiSE integrates intrinsic, ligand-induced, and protein-induced conformational changes within a single dataset constructed from native biological assemblies.

Using ProMiSE, we evaluate state-of-the-art models including AlphaFold3 and other recent generative models. We observe systematic conformational collapse: models struggle to sample intrinsic multi-states (∼20% cluster-level success), frequently converging to a single dominant state. In ligand-induced scenarios, predictions collapse toward holo-like conformations even under apo-conditioned inputs.

To understand these failures, we trace the prediction behavior through the model pipeline—training data composition, input multiple sequence alignment (MSA), intermediate pair representations, and final output structures. Through controlled analysis, we identify that the primary bottleneck of the failure mostly lies in the structure module rather than pair representation diversity. Training set memorization can constrain the structure module to collapse pair representations toward dominant conformations, even when

---

[*]Equal contribution [1]Department of Chemistry, Seoul National University, Gwanak-ro, Gwanak-gu, Seoul, Republic of Korea [2]Korea Institute of Science and Technology (KIST), Seoul, Republic of Korea. Correspondence to: Hahnbeom Park <hahnbeom@kist.re.kr>, Chaok Seok <chaok@snu.ac.kr>.

*Proceedings of the 43$^{rd}$ International Conference on Machine Learning*, Seoul, South Korea. PMLR 306, 2026. Copyright 2026 by the author(s).

distograms show no clear state preference.

Beyond providing a curated dataset, ProMiSE establishes the first unified evaluation framework that defines explicit success criteria for multi-state sampling and introduces quantitative metrics for assessing conformational bias across distinct biological contexts. We anticipate that this benchmark will serve as a standardized foundation for evaluating and developing next-generation, dynamics-aware protein structure prediction models.

**Conflict of Interest Disclosure.** The authors declare no competing interests.

## 2. Methods

### 2.1. Dataset Curation

#### 2.1.1. RETRIEVING MULTI-STATE PAIRS

To identify proteins exhibiting structural diversity, we retrieved sequences from the RCSB PDB grouped into clusters with $\geq 95\%$ sequence identity. For each sequence cluster, we performed all-against-all pairwise TM-score calculations and grouped entries into conformational clusters via agglomerative clustering. For sequence clusters with multiple conformational clusters, we selected pairs of assemblies from different conformational clusters with TM-score $< 0.8$, yielding 1,333,459 initial pairs.

#### 2.1.2. FILTERING MULTI-STATE PAIRS

To ensure structural integrity and biological relevance, we applied filtering criteria including crystal artifact removal (Elez et al., 2018), redundancy elimination, per-cluster sequence representative selection, and quality controls (see Appendix A for details).

#### 2.1.3. CATEGORIZING PAIRS BY CONFORMATIONAL CHANGE MECHANISM

Using the above pipeline, we collected conformations for each protein and then selected conformational pairs exhibiting structural transitions (TM-score $< 0.8$). We categorized these pairs into three mechanistic categories based on the differences between the two structures:

1. **Intrinsic Multi-State**: Both structures are monomeric and ligand-free, capturing conformational changes inherent to the protein sequence itself.

2. **Ligand-induced**: In this study, we refer to ligands as small-molecule ligands only. One structure in these pairs is monomeric and ligand-free (apo), while the other is ligand-bound (holo), capturing conformational changes triggered by small-molecule binding.

3. **Protein-induced**: One structure is monomeric and ligand-free (apo), while the other is a multimeric assembly (holo, $\geq 2$ protein chains). To isolate the effect of protein–protein interactions, we required that the small-molecule ligand profile of the target chain remain identical between the two structures.

For consistency, we refer to the binding-partner–free structure as apo, and the bound structure (ligand-bound or multimeric) as holo. In the case of intrinsic multi-state, both structures are ligand-free monomers, and thus a biologically meaningful apo/holo distinction does not exist. Nevertheless, downstream analyses such as bias quantification require assigning each structure to a single, discrete conformation state. For intrinsic multi-state pairs, we therefore assign apo and holo labels arbitrarily to maintain a unified evaluation framework, without implying any biological asymmetry.

*Table 1.* Summary statistics of the ProMiSE dataset.

| Category | Seq. Clusters | Entries (Apo / Holo) | Pairs |
|---|---|---|---|
| Intrinsic Multi-State | 72 | 162 (162 / —) | 109 |
| Ligand-induced | 87 | 435 (92 / 343) | 350 |
| Protein-induced | 75 | 189 (81 / 108) | 118 |
| Total | 234 | 786 | 577 |

Table 1 summarizes the final dataset composition, reporting the number of **sequence clusters** (non-redundant protein families at 40% sequence identity), **entries** (unique PDB structures), and **pairs** (entry combinations with TM-score $< 0.8$). For the intrinsic multi-state category, the dataset contains 72 clusters, with 58 two-state systems and 14 systems with three or more states.

### 2.2. Model Inference

Structures were predicted for all three dataset categories using AlphaFold3 (AF3) (Abramson et al., 2024), Boltz-1 (Wohlwend et al., 2025), Boltz-2 (Passaro et al., 2025), Chai-1 (Chai Discovery team et al., 2024), and BioEmu (Lewis et al., 2025). BioEmu was applied only to the intrinsic multi-state set.

For all predictions, the representative sequence from each sequence-identity cluster was used for target chain input sequence (see Appendix A for details). Each structure was predicted with 10 independent random seeds, generating 10 samples per seed, yielding 100 structural predictions per entry. MSAs were retrieved from the ColabFold server (Mirdita et al., 2022) using MMseqs2 (Steinegger & Söding, 2017). For multimeric inputs, MSA pairing was performed using the internal pairing algorithm of Boltz-2. Chai-1 predictions were run without MSA input.

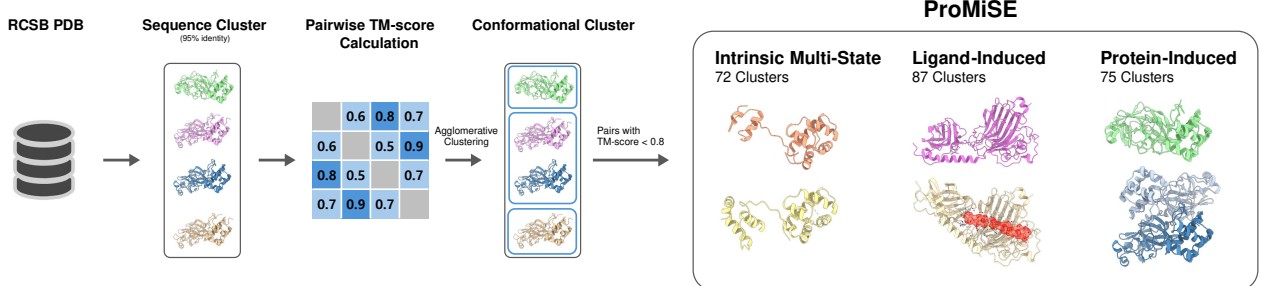

*Figure 1.* Overview of the ProMiSE dataset curation pipeline. Sequence clusters at 95% identity were retrieved from RCSB, followed by conformational clustering based on pairwise TM-score calculations. Structural pairs with TM-scores below 0.8 and belonging to different conformational clusters were selected. After further refinement, three types of entry pairs were extracted according to biological context: intrinsic multi-state, ligand-induced, and protein-induced.

## 2.3. Evaluation Metrics

### 2.3.1. STRUCTURE ALIGNMENT PROTOCOL

To assess monomeric conformation accuracy, target chains were extracted from both apo-conditioned and holo-conditioned predictions and aligned to experimental target chains using UCSF Chimera's MatchMaker algorithm (Meng et al., 2006) with the default option.

Each prediction was aligned not only to its corresponding experimental target but also to all other conformational states within the pair (e.g., apo-conditioned prediction to holo target), enabling assessment of conformational bias (i.e., $\text{Struct}_{\text{holo}}$ score defined in Section 2.4.1). For each alignment, C$\alpha$-RMSD and TM-score were calculated over aligned regions, excluding disordered or unresolved segments.

### 2.3.2. CONFORMATION SAMPLING SUCCESS

We define successful conformational sampling based on the dataset category.

**Intrinsic Multi-State.** For each conformational state, *state-level success* is defined as: a prediction with a TM-score $> 0.8$ to the target state and higher than to all other states. *Cluster-level success* is defined as: among 100 sampled predictions, at least one *state-level success* for each known conformational state in the cluster.

**Induced Sets.** For the induced sets, success is evaluated at the pair level and requires both apo and holo success. Let $P_{\text{apo}}$ and $P_{\text{holo}}$ denote apo-conditioned and holo-conditioned predictions, respectively. Let $\mathcal{A}$ denote the set of apo reference structures and $h$ denote the holo reference structure. For a prediction $P$, we define:

$$\text{TM}_{\text{apo}}(P) = \max_{a \in \mathcal{A}} \text{TM}(P, a), \qquad \text{TM}_{\text{holo}}(P) = \text{TM}(P, h).$$

This definition accounts for entries with multiple apo structures by using the best-matching apo reference for each prediction.

Apo success:

$$\text{TM}_{\text{apo}}(P_{\text{apo}}) > 0.8 \ \wedge \ \text{TM}_{\text{apo}}(P_{\text{apo}}) > \text{TM}_{\text{holo}}(P_{\text{apo}})$$

Holo success:

$$\text{TM}_{\text{holo}}(P_{\text{holo}}) > 0.8 \ \wedge \ \text{TM}_{\text{holo}}(P_{\text{holo}}) > \text{TM}_{\text{apo}}(P_{\text{holo}})$$

## 2.4. Conformational Bias Quantification

Modern protein structure prediction models follow a staged architecture in which evolutionary information derived from MSAs or language model outputs is processed into single and pairwise representations, refined through modules such as the Evoformer or Pairformer, and finally mapped to three-dimensional coordinates by a structure module (Jumper et al., 2021). Conformational bias toward apo or holo states may therefore arise at multiple stages of this pipeline, as well as from external factors such as training data bias.

To systematically analyze such conformational preferences, we introduce a set of complementary metrics that probe distinct stages of the modeling process and external data sources. Specifically, we quantify (i) bias in the final predicted structures, (ii) bias in internal pairwise distance predictions prior to structure generation, using predicted distograms as a proxy for pair representations, and external bias from (iii) the training set and (iv) MSA-derived coevolutionary signals. For all metrics defined below, positive values indicate holo preference and negative values indicate apo preference (e.g., $\text{Struct}_{\text{holo}} > 0$ indicates that the predicted structure is closer to the holo state, while $\text{Train}_{\text{holo}} < 0$ indicates a greater prevalence of apo-like examples in the training set). Chai-1 handles MSA differently from other models and does not provide pair representations. It was therefore run without MSA input and excluded from the conformational bias analysis.

### 2.4.1. STRUCTURE HOLO PREFERENCE SCORE (STRUCT$_{\text{HOLO}}$)

To quantify the structural bias of model predictions toward a specific conformational state, we adopt the ConfBench score from NeuralPLexer3 (Qiao et al., 2025):

$$\text{Struct}_{\text{holo}} = \frac{d_{\text{apo}} - d_{\text{holo}}}{\sqrt{0.5 \left( d_{\text{apo}}^2 + d_{\text{holo}}^2 + d_{\text{ref}}^2 \right)}}. \tag{1}$$

with $d_{\text{apo}} = \text{RMSD}(x_{\text{pred}}, x_{\text{apo}})$, $d_{\text{holo}} = \text{RMSD}(x_{\text{pred}}, x_{\text{holo}})$, $d_{\text{ref}} = \text{RMSD}(x_{\text{apo}}, x_{\text{holo}})$, where $x_{\text{pred}}$, $x_{\text{apo}}$, and $x_{\text{holo}}$ denote the predicted model, apo target, and holo target structures, respectively.

### 2.4.2. DYNAMIC REGION DISTOGRAM HOLO PREFERENCE SCORE (DYNDISTO$_{\text{HOLO}}$)

To quantify the model's internal structural bias at the distogram level (prior to the structure module), we first define a pair-wise expected distance error for each residue pair $(i, j)$. Let $p_{ij}(d)$ denote the predicted distance distribution from the distogram for residue pair $(i, j)$, and let $d_{\text{state}}^{(ij)}$ be the true inter-residue distance in a given conformational state (apo or holo). The expected absolute distance error for a single residue pair is defined as:

$$E_{\text{state}}^{(ij)} = \mathbb{E}_{d \sim p_{ij}(d)} \left[ |d - d_{\text{state}}^{(ij)}| \right] = \int |d - d_{\text{state}}^{(ij)}| \, p_{ij}(d) \, \mathrm{d}d. \tag{2}$$

Here, $d_{\text{state}}^{(ij)}$ is measured as the $C_\beta$–$C_\beta$ distance in the experimental structure for the corresponding state, with $C_\alpha$ used for glycine residues.

To specifically assess conformational bias in structurally dynamic regions, we restrict the analysis to residue pairs whose inter-residue distances differ substantially between the apo and holo states. We define the set of dynamic residue pairs as:

$$\mathcal{P}_{\text{dyn}} = \left\{ (i, j) \; : \; \left| d_{\text{apo}}^{(ij)} - d_{\text{holo}}^{(ij)} \right| > 3 \, \text{Å} \right\}. \tag{3}$$

The expected distance error in dynamic regions for a given state is then defined as:

$$E_{\text{state}}^{\text{dyn}} = \frac{1}{|\mathcal{P}_{\text{dyn}}|} \sum_{(i,j) \in \mathcal{P}_{\text{dyn}}} E_{\text{state}}^{(ij)}. \tag{4}$$

Using these quantities, the dynamic-region distogram holo preference score (DynDisto$_{\text{holo}}$) is computed by applying the same normalization as Eq. (1) to $E_{\text{apo}}^{\text{dyn}}$, $E_{\text{holo}}^{\text{dyn}}$, and $E_{\text{ref}}^{\text{dyn}}$, where

$$E_{\text{ref}}^{\text{dyn}} = \frac{1}{|\mathcal{P}_{\text{dyn}}|} \sum_{(i,j) \in \mathcal{P}_{\text{dyn}}} \left| d_{\text{apo}}^{(ij)} - d_{\text{holo}}^{(ij)} \right|. \tag{5}$$

quantifies the average structural difference between the apo and holo states within dynamic regions. The dynamic-region distogram holo preference score is then defined as:

$$\text{DynDisto}_{\text{holo}} = \frac{E_{\text{apo}}^{\text{dyn}} - E_{\text{holo}}^{\text{dyn}}}{\sqrt{0.5 \left( E_{\text{apo}}^{\text{dyn}\,2} + E_{\text{holo}}^{\text{dyn}\,2} + E_{\text{ref}}^{\text{dyn}\,2} \right)}}. \tag{6}$$

### 2.4.3. TRAINING SET HOLO SKEWEDNESS (TRAIN$_{\text{HOLO}}$)

To quantify how strongly the training set favors each conformational state, we estimated model-specific training exposure for the apo and holo states. Effective training hits were computed within the model-accessible training search space, incorporating the curation filters and weighted sampling scheme used during training. For consistency across AF3, Boltz-1, and Boltz-2, we used a common effective-hit formulation based on the reported AF3-style training curation and sampling scheme.

For each state $s \in \{\text{apo}, \text{holo}\}$, similar training entries were defined as chain- or interface-level entries with sequence identity $> 0.8$ and TM-score $> 0.9$ to the query state, and were searched with MMseqs2 and Foldseek (Van Kempen et al., 2024). The effective training hit count for state $s$, denoted $H_s^{\text{train}}$, was computed as the sum of training weights over these similar entries:

$$H_s^{\text{train}} = \sum_{i \in \mathcal{S}(s)} \frac{\beta_{r_i}}{N_{\text{clust},i}} \left( \alpha_{\text{prot}} n_{\text{prot},i} + \alpha_{\text{ligand}} n_{\text{ligand},i} \right), \tag{7}$$

where $\mathcal{S}(s)$ denotes the set of training entries similar to state $s$, $N_{\text{clust},i}$ is the cluster size of entry $i$, and $\beta_{r_i}$ is the item-type-specific sampling weight for chains or interfaces. Following the AlphaFold3 training regime, we used $\beta_{\text{chain}} = 0.5$, $\beta_{\text{interface}} = 1$, $\alpha_{\text{prot}} = 3$, and $\alpha_{\text{ligand}} = 1$.

For each apo–holo pair, we then defined the training-set holo skewedness as:

$$\text{Train}_{\text{holo}} = \frac{H_{\text{holo}}^{\text{train}} - H_{\text{apo}}^{\text{train}}}{H_{\text{holo}}^{\text{train}} + H_{\text{apo}}^{\text{train}}}. \tag{8}$$

Thus, positive Train$_{\text{holo}}$ values indicate greater training exposure to holo-like structures, whereas negative values indicate greater exposure to apo-like structures.

### 2.4.4. MSA HOLO PREFERENCE SCORE (MSA$_{\text{HOLO}}$)

To quantify the bias of MSA coevolutionary signals toward specific conformational states, we define:

$$\text{MSA}_{\text{holo}} = \frac{\text{MSA}_{\text{overlap}}^{\text{holo}} - \text{MSA}_{\text{overlap}}^{\text{apo}}}{\text{MSA}_{\text{overlap}}^{\text{holo}} + \text{MSA}_{\text{overlap}}^{\text{apo}}}. \tag{9}$$

where $\text{MSA}_{\text{overlap}}^{\text{state}}$ quantifies how strongly the MSA coevolutionary signal supports state-specific structural contacts. State-unique contacts are defined as residue pairs in contact (heavy atom distance $< 8\text{Å}$, excluding sequence neighbors $|i - j| \leq 3$) in one state but not in the other. The MSA overlap is computed as (Lee et al., 2025):

$$\text{MSA}_{\text{overlap}}^{\text{state-unique}} = \sum_{i<j} A_{ij} \cdot C_{ij}^{\text{state-unique}}, \qquad (10)$$

where $A_{ij}$ is the MSA attention score from the MSA Transformer (Rao et al., 2021), and $C_{ij}^{\text{state-unique}}$ is a binary contact indicator (1 if residues $i$ and $j$ are in contact exclusively in that state).

### 2.5. Data and Code Availability

The ProMiSE benchmark dataset and associated code are publicly available at https://github.com/seoklab/promise-bench.

## 3. Results & Discussion

### 3.1. Comparative Performance of Multiple Models

The benchmarking results reveal distinct performance profiles across the three mechanistic categories (Figure 2A). AF3 achieves the highest performance in ligand-induced transitions (0.50), while showing moderate accuracy for protein-induced (0.27) changes. Chai-1 shows balanced performance across categories despite its MSA-free setting, achieving competitive results in ligand-induced (0.40) and intrinsic multi-state (0.24), though it underperforms in protein-induced changes (0.17).

Boltz-2 exhibits a high success rate in protein-induced conformational changes, leading all models in this category (0.34), and also performing well for ligand-induced transitions (0.42). Both AF3 and Boltz-2, however, show limited accuracy in intrinsic multi-state (0.18 each). BioEmu performs best on intrinsic multi-state (0.29). In contrast, Boltz-1 underperforms across most categories, with the lowest success rate in the intrinsic multi-state set.

### 3.2. Limited Multi-State Sampling in Current Models

We evaluated five models on ProMiSE, revealing systematic conformational sampling limitations across all three mechanistic categories (Figure 2).

**Intrinsic Multi-State.** In the intrinsic multi-state set, all models demonstrate limited sampling capacity (Figure 2A). Although BioEmu achieves the highest cluster-level success rate at 29%, other models remain modest (8%–24%), indicating that even the best-performing model fails to adequately sample the conformational repertoire in over 70% of clusters.

Analysis of failure modes (Figure 2B) reveals three categories: *one-state collapse*, where all 100 predictions converge to a single state; *partial collapse*, where only a subset of states is sampled; and *complete failure*, where none of the known states are recovered. Conformational collapse (i.e., one-state and partial collapse) emerges as the dominant failure mode, indicating that failures are driven by systematic bias rather than random prediction errors. A representative example (Figure 2D, left) illustrates this behavior, where Boltz-2 predictions concentrate on a single dominant state.

Because our evaluation is existence-based, it is designed to test whether a model can recover each experimentally observed conformational state at least once, rather than whether it reproduces the full conformational ensemble or relative state populations. This distinction is important because estimating the true Boltzmann distribution of protein conformations remains beyond the scope of this benchmark. Nevertheless, we further examined whether, once a state is recovered, models allocate a substantial fraction of samples to experimentally observed states. Within successfully predicted intrinsic multi-state clusters, the hit rate—the fraction of samples matching at least one ground-truth state—remains substantial across models (mean 0.6–0.8; Table S1). This suggests that the primary limitation is not merely insufficient local sampling around already recovered states, but rather a failure to discover all relevant conformational states. Moreover, model confidence scores do not provide a reliable solution to this discovery problem: confidence-based selection fails to outperform uniform random sampling in cluster-level success rate (Figure S2), indicating that current confidence estimates are poorly calibrated for identifying alternative experimentally observed states.

**Ligand-Induced set.** For ligand-induced pairs, models show moderate overall success rates but exhibit directional bias (Figure 2A,C). Examination of $\text{Struct}_{\text{holo}}$ score distributions reveals that apo-conditioned predictions (blue boxes) systematically shift toward positive values, indicating collapse to holo-like conformations even in the absence of ligands. In contrast, holo-conditioned predictions (red boxes) appropriately favor holo conformations. This asymmetry demonstrates that the majority of failures of the ligand-induced set in Figure 2A arise from the inability to predict apo conformations, rather than from difficulties in predicting holo conformations when ligands are present (Figure S5). A representative case (Figure 2D, right) illustrates this holo collapse where the AF3 apo-conditioned prediction adopts a holo-like conformation despite the absence of ligand.

**Protein-Induced set.** In the protein-induced set, the directional bias observed in ligand-induced pairs is substantially reduced (Figure 2C, right panel). Both apo-conditioned and holo-conditioned predictions display relatively bal-

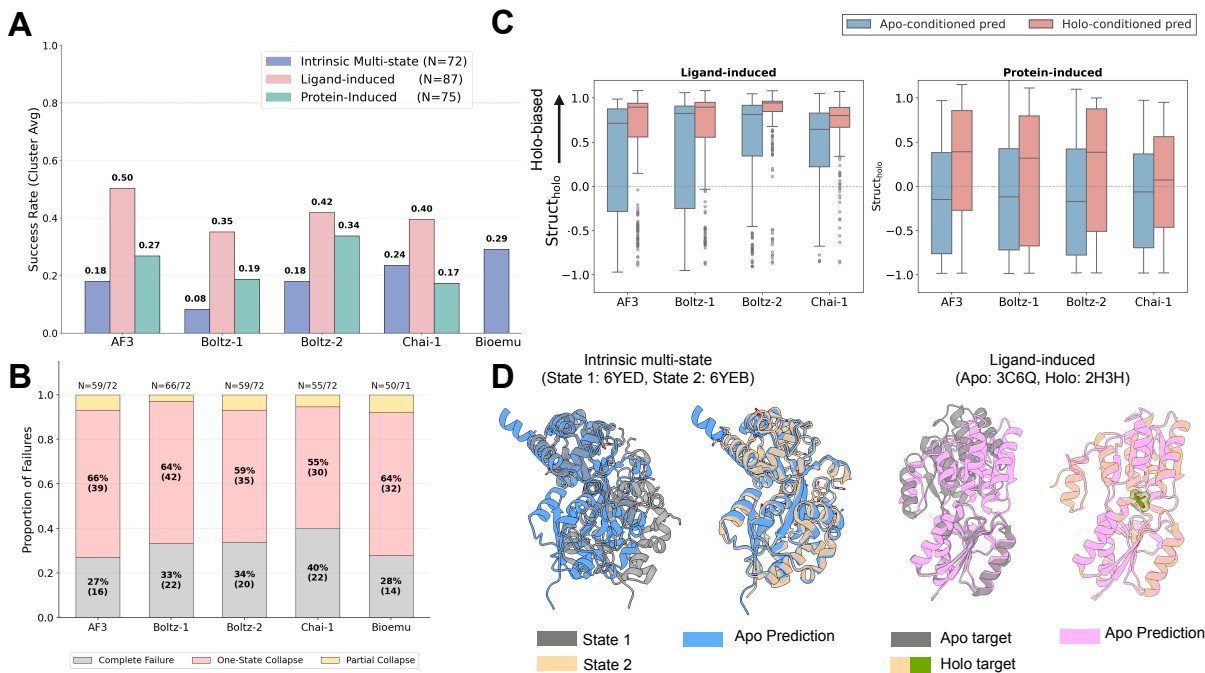

*Figure 2.* Conformational sampling performance across models. (A) Cluster-level success rates across three dataset categories. (B) Failure mode breakdown for intrinsic multi-state. (C) $Struct_{holo}$ score distributions for apo-conditioned (blue) and holo-conditioned (red) predictions in ligand-induced (left) and protein-induced (right) sets. Positive values indicate holo-like conformations, negative values indicate apo-like conformations. (D) Structural examples showing conformational collapse. *Left*: intrinsic multi-state (PDB 6YED, gray; PDB 6YEB, tan) with Boltz-2 predictions (blue), where the prediction closest to state 1 was selected. *Right*: Ligand-induced apo target (PDB 3C6Q, gray), holo target (PDB 2H3H, tan), and AF3 apo-conditioned prediction (pink), where the prediction closest to the apo state was selected.

anced $Struct_{holo}$ distributions, with apo-conditioned predictions tending toward negative values (apo-like) and holo-conditioned predictions toward positive values (holo-like). However, despite this apparent balance, the overall success rate remains low (Figure 2A). This discrepancy arises because many predictions cluster near $Struct_{holo} \approx 0$, which by definition indicates structures that are equally distant from both apo and holo references—neither successfully recovering the apo conformation nor the holo conformation. Thus, unlike the systematic holo collapse in ligand-induced pairs, models in the protein-induced set fail to sample either state, producing structures that match neither functional state.

**Conformational Collapse.** Accurate conformational sampling of individual monomers is critical for overall complex structure prediction accuracy, as demonstrated by the correlation between monomer conformational accuracy and full complex accuracy in induced sets (Figure S3). However, both intrinsic multi-state and ligand-induced sets exhibit systematic bias toward a single state—a phenomenon we term ***conformational collapse***. In the following sections, we investigate the origins of this conformational collapse by analyzing biases at different stages of the prediction

pipeline.

### 3.3. Training Set Bias and Limited Conformational Sampling

Given the limitations of multi-state sampling observed above, we examine where conformational bias can emerge within the model architecture and which factors are associated with this behavior.

#### 3.3.1. OBSERVING WHERE BIAS ARISES IN THE MODEL: PAIR REPRESENTATION VS. STRUCTURE MODULE

To localize where conformational preferences arise during inference, we compared bias at two stages of prediction: the distogram output ($DynDisto_{holo}$) and the structure prediction output ($Struct_{holo}$). These quantities capture bias before and after the structure module, respectively, allowing us to assess whether conformational preferences are already present in pairwise representations or emerge within the structure module itself.

We evaluated these metrics separately for apo-conditioned and holo-conditioned predictions (Figure S4). In this dis-

cussion, we focus on the ligand-induced set, where conformational collapse is most severe and overwhelmingly directional ($\geq$80% of apo-conditioned failures correspond to Apo $\rightarrow$Holo collapse across models; Figure S5).

In the ligand-induced set, we observe a partial dissociation between distogram-level and structure-level preferences. At the distogram stage, predictions retain some sensitivity to their input context relative to the structure module: apo-conditioned predictions lean toward apo and holo-conditioned predictions toward holo, although AF3 and Boltz-1 show a mild holo bias even under apo conditioning (Figure S4). At the structure module output, however, both apo- and holo-conditioned predictions converge toward positive $\text{Struct}_{\text{holo}}$ values, indicating that the holo shift is markedly stronger after the structure module than at the distogram stage.

This collapse indicates that conformational bias is not solely introduced in the pair representations, but is further amplified during the structure module. Notably, for apo-conditioned predictions where the distogram does not exhibit extreme conformational preference, the structure module tends to shift predictions toward holo-like conformations, with this effect being most pronounced in the Boltz models and comparatively weaker in AF3 (Figure S8). This suggests that the structure module preferentially collapses pairwise representations without a strong distogram-level state preference toward holo-biased conformations, even for apo-conditioned predictions, rather than maintaining conformational diversity.

### 3.3.2. TRAINING SET MEMORIZATION DRIVES STRUCTURE MODULE BIAS

To identify factors contributing to the observed conformational collapse in the structure module, we considered two potential drivers based on prior literature (Lee et al., 2025; Wayment-Steele et al., 2024): (1) MSA coevolutionary signals ($\text{MSA}_{\text{holo}}$) inherently favoring one conformational state, and (2) training set skewedness ($\text{Train}_{\text{holo}}$) causing the structure module to memorize dominant training conformations.

To disentangle these factors, we performed an analysis in which one source of bias was held neutral (low magnitude) while the effect of the other was examined (Figure S8, S9). Specifically, we analyzed the relationship between $\text{DynDisto}_{\text{holo}}$ and $\text{Struct}_{\text{holo}}$ under two complementary conditions. First, we restricted the analysis to pairs with minimal training set skewedness ($|\text{Train}_{\text{holo}}| < 0.3$) and visualized the results by coloring points according to $\text{MSA}_{\text{holo}}$ (Figure S9). Conversely, we selected pairs with weak MSA signal ($|\text{MSA}_{\text{holo}}| < 0.3$) and colored the corresponding points according to $\text{Train}_{\text{holo}}$ (Figure S8).

Figure 3A indicates that models' biased exposure to certain states during training is the major source of the bias emerging at the structure module. To test this, we measured the additional holo-state bias introduced after the distogram stage by defining shift as $\text{Struct}_{\text{holo}} - \text{DynDisto}_{\text{holo}}$ and computing the correlation with $\text{Train}_{\text{holo}}$ and $\text{MSA}_{\text{holo}}$, respectively. Across models, $\text{MSA}_{\text{holo}}$ showed little to no correlation with this shift, whereas $\text{Train}_{\text{holo}}$ exhibited a consistent positive correlation. These results suggest that the additional holo-state bias introduced after the distogram stage is attributable to bias in the training set, rather than to bias in the input MSA. In the ligand-induced holo-conditioned setting, both the distogram and final structure predictions were already strongly biased toward the holo state, which likely saturated the signal and made the corresponding correlation estimates noisy.

**Case study: An outlier to the memorization hypothesis.** Although the analysis above shows a strong correlation between $\text{Train}_{\text{holo}}$ and $\text{Struct}_{\text{holo}}$ under neutral distograms, we identified an outlier cluster (black circles in Figure 3A) that deviates from this trend. Despite strong holo bias at the chain level, AF3 predicted apo-like conformations, whereas Boltz-2 predicted holo-like conformations.

To investigate this discrepancy, we examined a representative pair from the cluster (PDB IDs: 6crf and 7xhq), which undergoes a domain rearrangement involving the SH2 region (Figure S7A). Because SH2 domains are highly conserved, chain-level training statistics may not adequately reflect the domain-level interaction patterns underlying this transition.

We therefore performed domain-level training set searches focusing on the domains directly involved in the motion (N-SH2 and PTP). Unlike chain-level analysis, this domain-restricted search revealed an apo bias ($\text{Train}_{\text{holo}}^{\text{domain}} = -0.1148$), reversing the direction of the inferred training-set preference (Figure S7B). Together, these results suggest that the observed outlier arises from differences in the structural resolution at which training-set memorization operates: AF3 appears more sensitive to domain-level statistics, whereas Boltz-2 aligns more closely with chain-level biases.

### 3.4. Validating the Hypothesis: BioEmu's Diverse Structure Decoding

The analyses in Section 3.3.1 suggest that, particularly in the intrinsic multi-state set, the structure module acts as the primary bottleneck for conformational diversity across all models. To test this hypothesis, we compared distogram-to-structure mapping behavior against BioEmu (Lewis et al., 2025), which uses the same AlphaFold2 pairformer (frozen) but fine-tunes only the structure decoding module on MD trajectory data.

**Intrinsic Multi-State Interpretation.** For the intrinsic

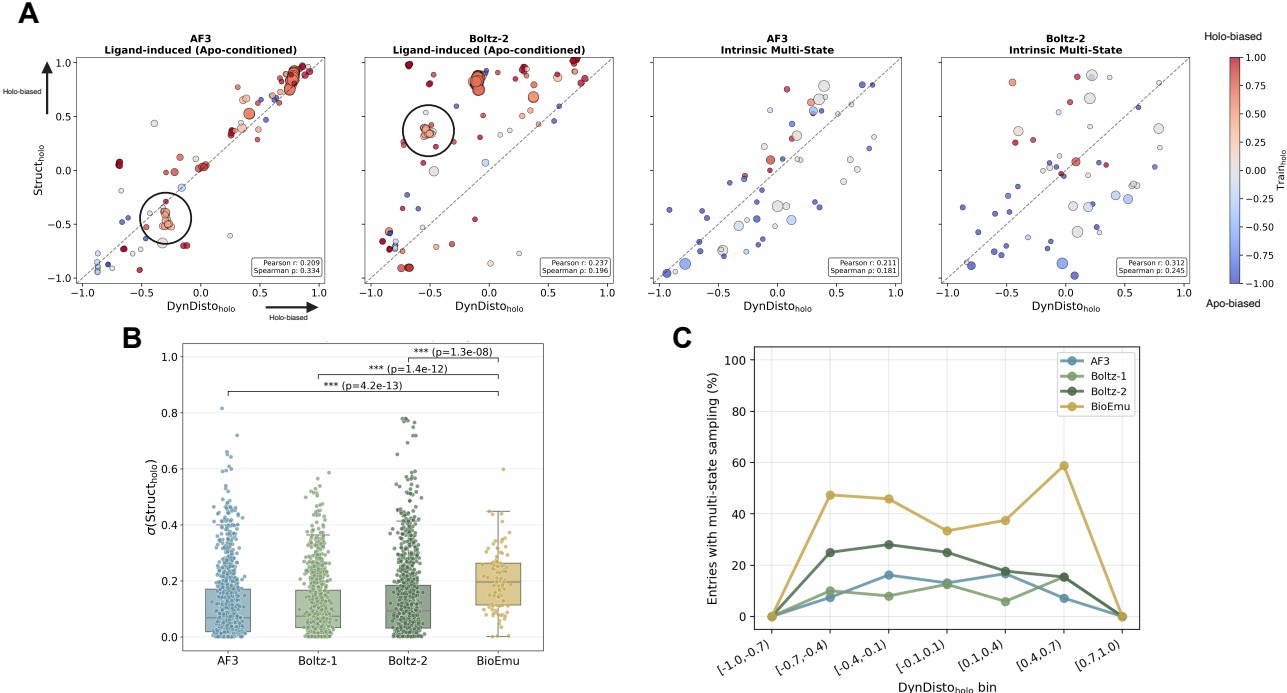

*Figure 3.* Distogram-level versus structure-level state preference across models. (A) Scatter plots of $DynDisto_{holo}$ ($x$-axis) versus $Struct_{holo}$ ($y$-axis). Points are colored by $Train_{holo}$ and sized according to the number of structure-based search hits. From left to right, the panels correspond to AF3 (ligand-induced set, apo-conditioned), Boltz-2 (ligand-induced set, apo-conditioned), AF3 (intrinsic multi-state), and Boltz-2 (intrinsic multi-state). The dashed line indicates $y = x$. Correlation coefficients (Pearson and Spearman) are computed between $Struct_{holo} - DynDisto_{holo}$ and $Train_{holo}$. (B) Within-distogram standard deviation of $Struct_{holo}$, where each dot represents one distogram. For AF3/Boltz, each of $\sim$10 seeds per target produces one dot, computed as the standard deviation of the 10 structure models within that seed (no subsampling). For BioEmu, which uses a single seed with 100 samples, 10 samples are subsampled and their standard deviation taken, repeated over 50 trials and averaged into one dot per target. Significance was assessed by paired Wilcoxon signed-rank tests comparing BioEmu against each of AF3, Boltz-1, and Boltz-2. (C) Fraction of entries where predictions span both conformational states (at least one $Struct_{holo} < -0.5$ and one $> +0.5$), stratified by $DynDisto_{holo}$ bin.

multi-state set, the apo/holo distinction is arbitrary, and thus directional bias such as *holo collapse* cannot be meaningfully defined. Nevertheless, conformational collapse is still apparent as a narrow range of structural outputs for a given distogram input.

**Collapse in existing models.** For AF3, Boltz-1, and Boltz-2, a given distogram maps to a narrow range of structural outputs: the within-distogram standard deviation of $Struct_{holo}$, $\sigma(Struct_{holo})$, is consistently low (Figure 3B). As a result, these models rarely sample both conformational states for a given target, with the fraction of multi-state entries remaining below $\sim$30% across all $DynDisto_{holo}$ bins (Figure 3C).

**BioEmu's diverse decoding.** In contrast, BioEmu produces substantially more diverse structural outputs from a given distogram, with significantly higher $\sigma(Struct_{holo})$ than all three models (Figure 3B; Wilcoxon signed-rank test,

$p < 10^{-4}$ for all comparisons). This diversity translates into markedly higher multi-state sampling, particularly in the central $DynDisto_{holo}$ bins where the distogram does not strongly favor either state (Figure 3C).

**Validation of the bottleneck hypothesis.** Critically, this enhanced diversity emerges without more diverse distogram outputs; BioEmu uses the same frozen pairformer as AlphaFold2. These results validate our hypothesis: the limitation lies in the structure module's inability to preserve conformational diversity during decoding, constrained by training set memorization. Fine-tuning with conformationally diverse data can restore the model's ability to map ambiguous representations to diverse structural ensembles.

### 3.5. Limitations

The biological contexts of conformational change often overlap, and some entries do not fit cleanly into a single category under our current annotation scheme. This is particularly challenging for multimeric assemblies with bound ligands, where protein-induced and ligand-induced effects can be coupled. We therefore used conservative category definitions in this work and leave the incorporation of multimeric assemblies with ligand contexts to future benchmark extensions.

Our pair-extraction criterion, TM-score $< 0.8$, is also stringent and preferentially captures large conformational changes. Consequently, localized but biologically important motions, such as GPCR TM6 displacement or transporter pocket rearrangements, may be excluded. A complementary criterion of $0.8 <$ TM-score $< 0.9$ with RMSD $> 3$ Å captured GPCR multi-state cases (e.g., ligand-induced: PDB IDs 7UL5 and 7WIG; protein-induced: 7UL5 and 7T10), suggesting a practical route for future expansion. We retained the stricter TM-score cutoff here due to computational and curation resource constraints.

### 4. Conclusion

We introduced ProMiSE, a benchmark for evaluating multi-state protein structure prediction across intrinsic multi-state, ligand-induced, and protein-induced conformational changes. Our evaluation revealed systematic limitations: models achieve only ~20% cluster-level success for intrinsic multi-state and exhibit strong holo bias in ligand-induced scenarios.

Through controlled analysis across the prediction pipeline, we identified that a key bottleneck lies in the structure decoding module rather than solely in the diversity of pairwise representations upstream. Training set memorization constrains the structure module to produce restricted conformational variation from uncertain pairwise representations, even when distograms show no clear state preference. Critically, comparison against BioEmu supports this hypothesis: substantially greater diversity in distogram-to-structure mapping emerges when only the structure module is fine-tuned on conformationally diverse data, without requiring more diverse pair representations.

These findings have direct implications for future model development. Rather than focusing solely on MSA manipulation or architectural changes to the trunk network, our results suggest that modifications at the structure module stage may provide a practical path toward more multi-state-aware protein modeling.

ProMiSE provides both a systematic evaluation framework for measuring progress and actionable insights for improv-

ing multi-state prediction. As the field moves beyond static structure accuracy toward modeling protein dynamics, our benchmark and findings establish a foundation for developing and evaluating the next generation of context-aware, dynamics-aware protein structure prediction models.

### Acknowledgments

We thank Nuri Jung for developing the initial codebase for large-scale conformational clustering and multi-state pair curation, and for providing fast and robust structure alignment and score computation through Nurikit.

This work was supported by a grant of the National Research Foundation of Korea (NRF) (RS-2024-00407331), a grant of the Institute of Information & Communications Technology Planning & Evaluation (IITP) (RS-2023-00220628), and a grant of the Korea-US Collaborative Research Fund (KUCRF) (RS-2024-00467483) funded by the Ministry of Science and ICT and Ministry of Health & Welfare, Republic of Korea.

### Impact Statement

This paper presents work whose goal is to advance the field of Machine Learning. There are many potential societal consequences of our work, none which we feel must be specifically highlighted here.

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

# A. Appendix

## A.1. Related Works

### A.1.1. CONFORMATIONAL SAMPLING LIMITATIONS IN STRUCTURE PREDICTION

Despite substantial advances in single-structure prediction, current models exhibit systematic limitations in capturing conformational diversity, achieving only approximately 35% success in recovering alternative folds of fold-switching proteins (Chakravarty et al., 2024; Schafer & Porter, 2025). The sources of this bias remain debated: Wayment-Steele et al. (2024) showed that MSA clustering can expose distinct coevolutionary patterns driving alternative fold predictions, while Chakravarty et al. (2024) and Lazou et al. (2024) found that training set memorization often overrides MSA-derived signals. Similar memorization effects have been observed in protein-ligand co-folding models (Masters et al., 2025).

### A.1.2. DIVERSE CONFORMATIONAL SAMPLING METHODS

To address limited conformational diversity, researchers have pursued two major directions. The first leverages molecular dynamics data, as exemplified by AlphaFlow (Jing et al., 2024) and BioEmu (Lewis et al., 2025), which fine-tune structure prediction models on MD simulations. The second improves sampling through architectural innovations such as diffusion models (Wang et al., 2024), inference-time methods including MSA dropout (Del Alamo et al., 2022; Richman et al., 2025), and context-aware methods like NeuralPLexer (Qiao et al., 2024; 2025) that condition on binding partners.

### A.1.3. BENCHMARKS FOR CONFORMATION CHANGE EVALUATION

Large-scale datasets have been developed for protein-protein interactions (Kovtun et al., 2024) and protein-ligand binding (Durairaj et al., 2024), while recent efforts have introduced benchmark sets for intrinsic multi-state (Lewis et al., 2025) and ligand-induced changes (Qiao et al., 2024). However, these resources typically focus on isolated conformational mechanisms and decompose biological assemblies into monomeric units. A unified benchmark integrating multiple conformational mechanisms in biologically realistic assemblies remains absent.

## A.2. Dataset Curation

### A.2.1. RETRIEVING MULTI-STATE PAIRS.

To identify proteins exhibiting structural diversity, we retrieved sequences from the RCSB Protein Data Bank (PDB) grouped into clusters with $\geq 95\%$ sequence identity. Each "entry" in these clusters represents a single protein chain, designated as the Chain of Interest (COI) within its biological assembly. Lists of chains and ligands per assembly were maintained throughout the filtering process and used for classifying induced changes.

For each sequence cluster, we performed pairwise TM-score calculation between all entries, incorporating the structural coordinates of all author-defined assemblies. These scores were subsequently used to group entries into conformational clusters via agglomerative clustering. Sequence clusters containing only a single conformational cluster were excluded, as they do not represent multi-state proteins.

From the remaining multi-state clusters, we selected pairs of protein assemblies that (1) belong to the same sequence cluster but different conformational clusters and (2) exhibit a TM-score $< 0.8$. In total, 1,333,459 multi-state pairs were retrieved. Prior to pair filtering, ligands in each assembly were curated using an exclusion list. Ligands located beyond a 5 Å distance cutoff from the protein were removed. Metal ions were retained only if they coordinated with two or more residues within 5 Å; all other metal ions were excluded.

### A.2.2. FILTERING MULTI-STATE PAIRS.

To ensure the structural integrity and biological relevance of the identified multi-state pairs, we applied a series of stringent filtering criteria as follows:

**Crystal Artifact Removal.** Multimeric assemblies identified as containing crystal packing artifacts were excluded, based on the classification from Algorithm 1.

**Redundancy Elimination.** Redundant assemblies were systematically removed according to the superset and equivalence criteria defined in Algorithm 2. Also, we excluded pairs that include homomeric chains sharing the same PDB ID within a

single conformational cluster.

**Sequence Representative.** For fair comparison across structures, sequence inputs were unified within each sequence-identity cluster by selecting a single representative sequence. Because multiple entries within a cluster can differ at residues forming the binding interface, naïve representative selection may introduce binding-site inconsistencies that compromise comparison fidelity. To address this, we defined the binding site as residues within 5 Å of the binding partner and identified binding-site mismatches across entries in the same cluster. The representative sequence was chosen as the entry with the largest support (highest count) that exhibited no binding-site mismatches relative to other cluster members. Any pair involving entries with binding-site mismatches was excluded from subsequent analyses.

**Sequence Similarity Removal.** To remove sequence redundancy, cluster representatives were re-clustered via MMseqs2 (0.4 identity threshold). We selected the final representatives based on the maximum number of conformational clusters, using the total entry count as a tie-breaker.

**Auxiliary Filters.** In addition to the primary criteria above, the following quality-control filters were applied to ensure data consistency and manageability:

- Structures with a resolution worse than 5.0 Å were excluded.

- Software-defined assemblies were excluded.

- Complexes containing RNA molecules were excluded to maintain a focus on protein-specific conformational changes.

- Assemblies with more than 12 protein chains or more than 12 ligands were excluded to limit structural complexity.

**Conformation Cluster Filters.** To avoid conformational redundancy within each sequence cluster, we applied entry selection criteria: for the intrinsic multi-state set, we retained only one entry per conformation label; for the ligand-induced and protein-induced sets, we allowed multiple entries with the same conformation label if they differ in their binding partners. In addition, we discarded sequence clusters in which apo and holo entries were assigned to the same conformation cluster, as this indicates that the two states are structurally similar and lack a detectable induction effect.

---

**Algorithm 1** Identification and Removal of Crystal Artifacts

---

**Require:** Protein assembly $A$, Chain of Interest $C$, distance threshold $d_{min}$, PRODIGY-cryst threshold $T = 0.5$
**Ensure:** Updated assembly set (excluding artifacts)
 1: **Initialize** graph $G(V, E)$, where $V$ represents protein chains and $E$ represents contacts.
 2: **for** each edge $e_{ij} \in E$ between chain $i$ and $j$ **do**
 3:    **if** distance($C\alpha_i, C\alpha_j) > d_{min}$ **then**
 4:       $E \leftarrow E \setminus \{e_{ij}\}$ {Remove disconnected edges}
 5:    **else if** ligand exists in proximity to both $i$ and $j$ **then**
 6:       **continue** {Preserve ligand-mediated biological contacts}
 7:    **else**
 8:       $S_{ij} \leftarrow$ PRODIGY-cryst($e_{ij}$)
 9:       **if** $S_{ij} > T$ **then**
10:          $E \leftarrow E \setminus \{e_{ij}\}$ {Sever artifactual crystal contacts}
11:       **end if**
12:    **end if**
13: **end for**
14: $V_{sub} \leftarrow$ DFS($G, C$) {Find reachable chains from COI}
15: **if** $V_{sub} \neq V$ **then**
16:    **return** Artifact identified (Remove assembly)
17: **else**
18:    **return** Biological assembly (Keep assembly)
19: **end if**

---

---

**Algorithm 2** Assembly Redundancy Removal via Superset Detection

---

**Require:** Conformational cluster $C = \{A_1, A_2, \ldots, A_n\}$
**Ensure:** Non-redundant set of assemblies $C^*$
1: $C^* \leftarrow C$
2: **for** each pair $(A, B) \in C^* \times C^*$ where $A \neq B$ **do**
3: $is\_superset \leftarrow$ **true** {Check COI ligand equivalence}
4: **if** $\text{Ligands}(COI_A) \neq \text{Ligands}(COI_B)$ **or** $|\text{Neighbors}_A| \neq |\text{Neighbors}_B|$ **then**
5:  $is\_superset \leftarrow$ **false**
6: **else**
7:  Find matching $\sigma : \text{Neighbors}_B \rightarrow \text{Neighbors}_A$ {such that:}
8:  **for** each $n_i \in \text{Neighbors}_B$ **do**
9:   **if** $\text{SeqID}(n_i, \sigma(n_i)) < 0.95$ **or** $\text{Ligands}(n_i) \neq \text{Ligands}(\sigma(n_i))$ **then**
10:    $is\_superset \leftarrow$ **false**
11:    **break**
12:   **end if**
13:  **end for**
14: **end if**
15: **if** $is\_superset$ is **true then**
16:  **if** $|A| > |B|$ **then**
17:   $C^* \leftarrow C^* \setminus \{A\}$ {Remove the structural superset}
18:  **else if** $|A| = |B|$ **then**
19:   $A_{rem} \leftarrow \text{TieBreak}(A, B)$ {Apply resolution and PDB ID rules}
20:   $C^* \leftarrow C^* \setminus \{A_{rem}\}$
21:  **end if**
22: **end if**
23: **end for**
24: **return** $C^*$

---

## A.3. Supplementary Materials

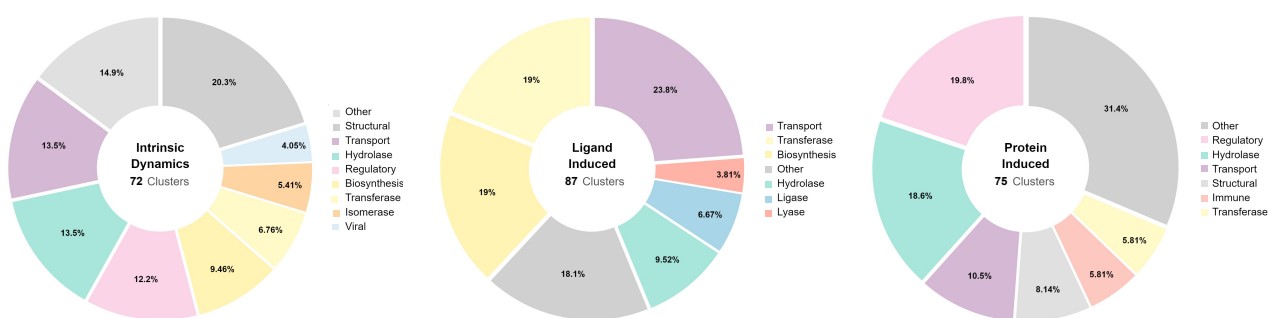

*Figure S1.* Statistics of functional groups in the ProMiSE dataset. Functional annotations were retrieved from Gene Ontology terms associated with each PDB entry. The resulting distribution indicates that the dataset spans a broad range of functional categories without over-representation of redundant protein groups, supporting the robustness of the dataset curation procedure.

*Table S1.* Hit rate within successful intrinsic multi-state clusters as a measure of partial distributional coverage. Values report the mean fraction of samples achieving TM-score $\geq 0.8$ to at least one ground-truth state, computed only over clusters where at least one ground-truth state was successfully recovered. Standard deviation is across clusters; the number of such clusters per model is also reported. Best results are shown in bold.

| METRIC | AF3 | BOLTZ-1 | BOLTZ-2 | CHAI-1 | BIOEMU |
|---|---|---|---|---|---|
| MEAN FRACTION ($\uparrow$) | 0.818 | **0.840** | 0.617 | 0.804 | 0.767 |
| STD ($\downarrow$) | 0.320 | 0.302 | 0.371 | 0.290 | **0.234** |
| # CLUSTERS | 19 | 13 | 16 | 19 | 24 |

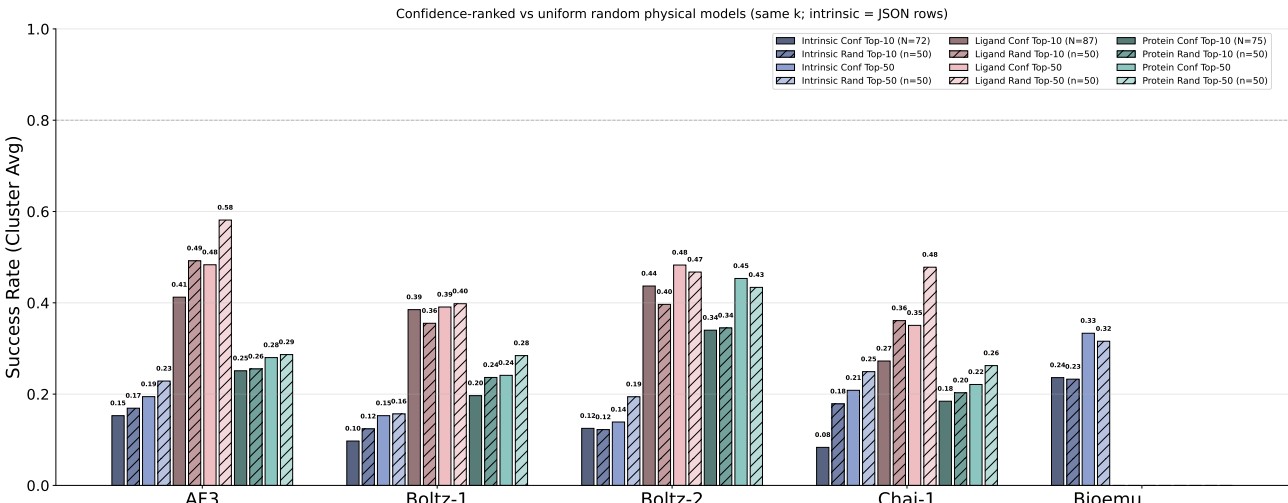

*Figure S2.* Comparison of confidence-based selection and uniform random sampling at matched sample budgets. Success rates are computed by selecting the top-$k$ samples based on model confidence scores and by uniformly sampling $k$ structures at random from the generated ensemble. Selecting predictions based on the model's top confidence scores (Confidence Top-10) yields cluster-level success rates that are nearly identical to, or occasionally outperformed by, uniform random sampling from the generated ensemble (Random Top-10 or Top-50). This indicates that current model confidence scores primarily reflect a preference for the memorized, dominant conformation rather than the biological validity of alternative functional states, making self-reported confidence a poor metric for evaluating multi-state sampling quality.

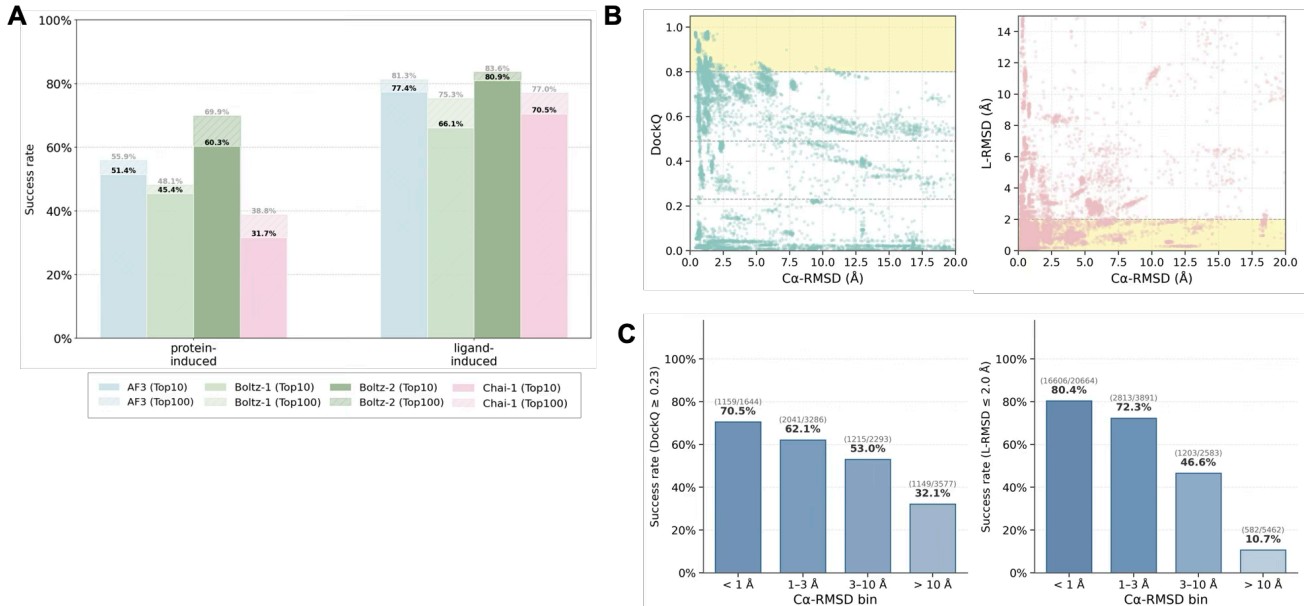

*Figure S3.* Monomer conformational accuracy correlates with complex structure prediction success. **(A)** Success rates for complex structure prediction in induced sets across models. Success is defined as DockQ $\geq 0.23$ for the protein-induced set and ligand RMSD $\leq 2$ Å for the ligand-induced set. **(B)** Correlation between monomer conformational accuracy (Cα-RMSD in Å) and complex structure accuracy for AF3 predictions. Left: protein-induced set, Cα-RMSD vs. DockQ (Pearson $r = -0.425$, Spearman $\rho = -0.426$, both $p \approx 0$), with the success region (DockQ $\geq 0.23$) shaded. Right: ligand-induced set, Cα-RMSD vs. ligand RMSD (Pearson $r = 0.342$, Spearman $\rho = 0.435$, both $p \approx 0$), with the success region (ligand RMSD $\leq 2.0$ Å) shaded. **(C)** Complex prediction success rate binned by monomer Cα-RMSD. Left: protein-induced set (DockQ $\geq 0.23$), declining from $70.5\%$ ($< 1$ Å) to $32.1\%$ ($> 10$ Å). Right: ligand-induced set (ligand RMSD $\leq 2.0$ Å), declining from $80.4\%$ ($< 1$ Å) to $10.7\%$ ($> 10$ Å). These results demonstrate that accurate monomer conformational sampling is critical for overall complex structure prediction accuracy.

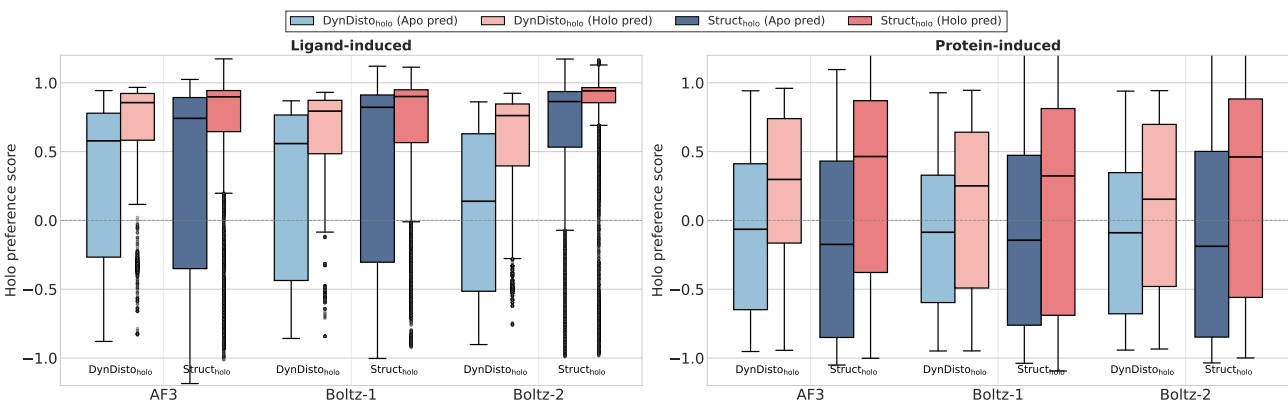

*Figure S4.* Comparison of distogram-based and structure-based holo preference scores across models. Boxplots show the distribution of distogram-based (DynDisto$_{holo}$) and structure-based (Struct$_{holo}$) holo preference scores for apo-conditioned and holo-conditioned predictions across AF3, Boltz-1, and Boltz-2. Left panel: ligand-induced set. Right panel: protein-induced set. For each model, scores are provided for both apo-conditioned and holo-conditioned inputs to compare conformational preferences at the pairwise representation stage (distogram) and final 3D structure stage (structure module output).

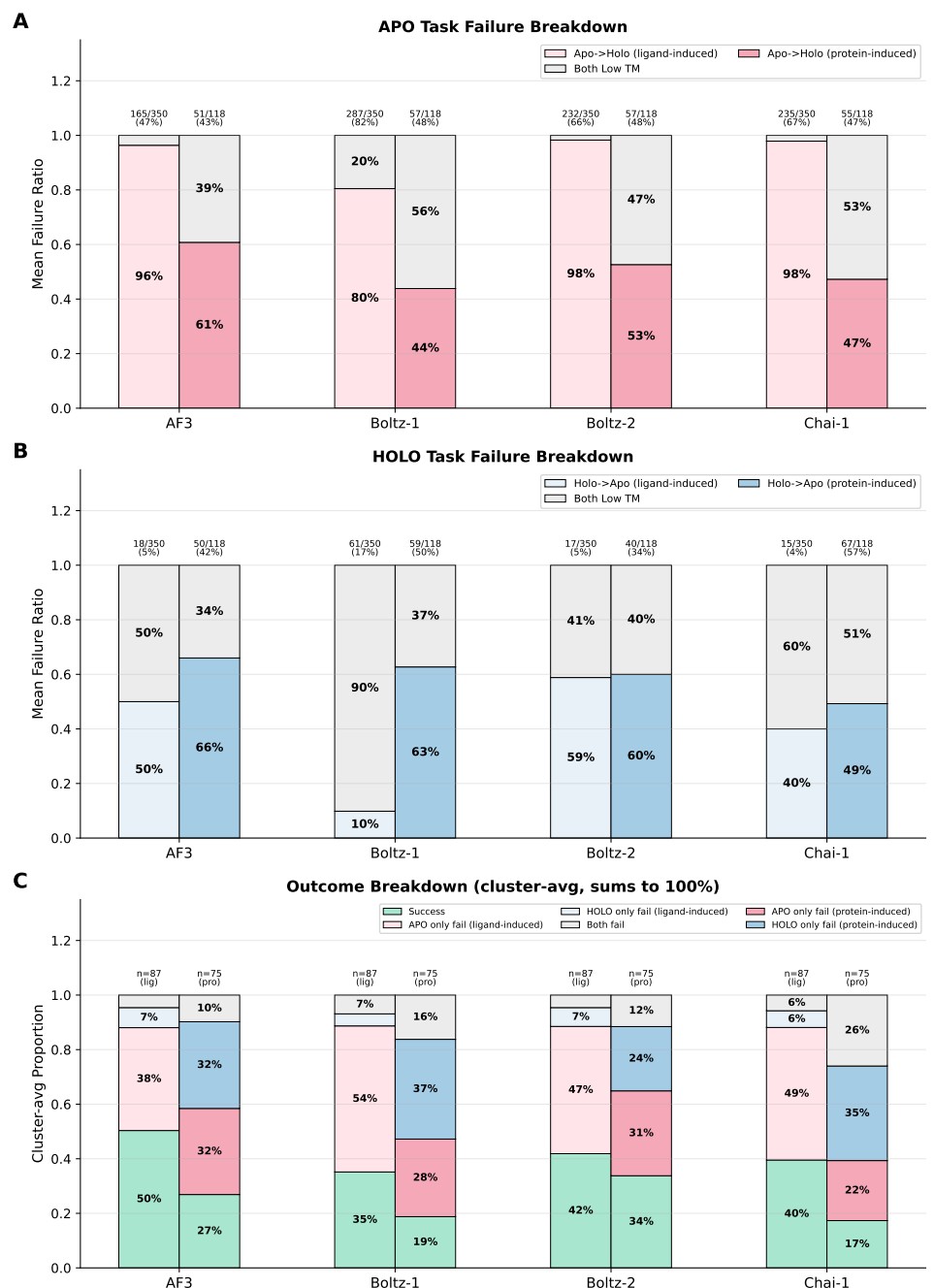

*Figure S5.* Failure-mode breakdown for ligand-induced and protein-induced sets. **(A, B)** Among entries where the apo- (A) or holo- (B) conditioned prediction failed, failures are partitioned into *directional collapse* – where the prediction matches the *other* reference well (TM $\geq$ 0.8) and is closer to it than to the target (e.g., for the apo task, $TM_{holo} \geq 0.8$ and $TM_{holo} > TM_{apo}$) – and *Both low TM*, where TM $< 0.8$ against both references. The fraction of failed entries among all entries is annotated above each bar. Apo failures in the ligand-induced set are overwhelmingly Apo→Holo directional collapses across all models, while protein-induced apo failures are roughly evenly split between directional and non-directional modes. Holo failures are rare in the ligand-induced set with no consistent directional bias, but are frequent in the protein-induced set with directional Holo→Apo collapse holding a slight majority. **(C)** Cluster-level outcome breakdown summing to 100% across *Success* (both apo and holo conditions succeed), *APO only fail*, *HOLO only fail*, and *Both fail*. Ligand-induced clusters (*lig*) are dominated by APO-only failure, consistent with the asymmetric Apo→Holo collapse in Section 3.2. Protein-induced clusters (*pro*) distribute failure across all three modes, with *HOLO only fail* or *APO only fail* as the largest share and *Both fail* consistently the smallest. Together, these breakdowns support the distinction that ligand-induced failures are *asymmetric* (collapse in a single direction, Apo→Holo), whereas protein-induced failures are *symmetric*, with substantial collapse in both directions.

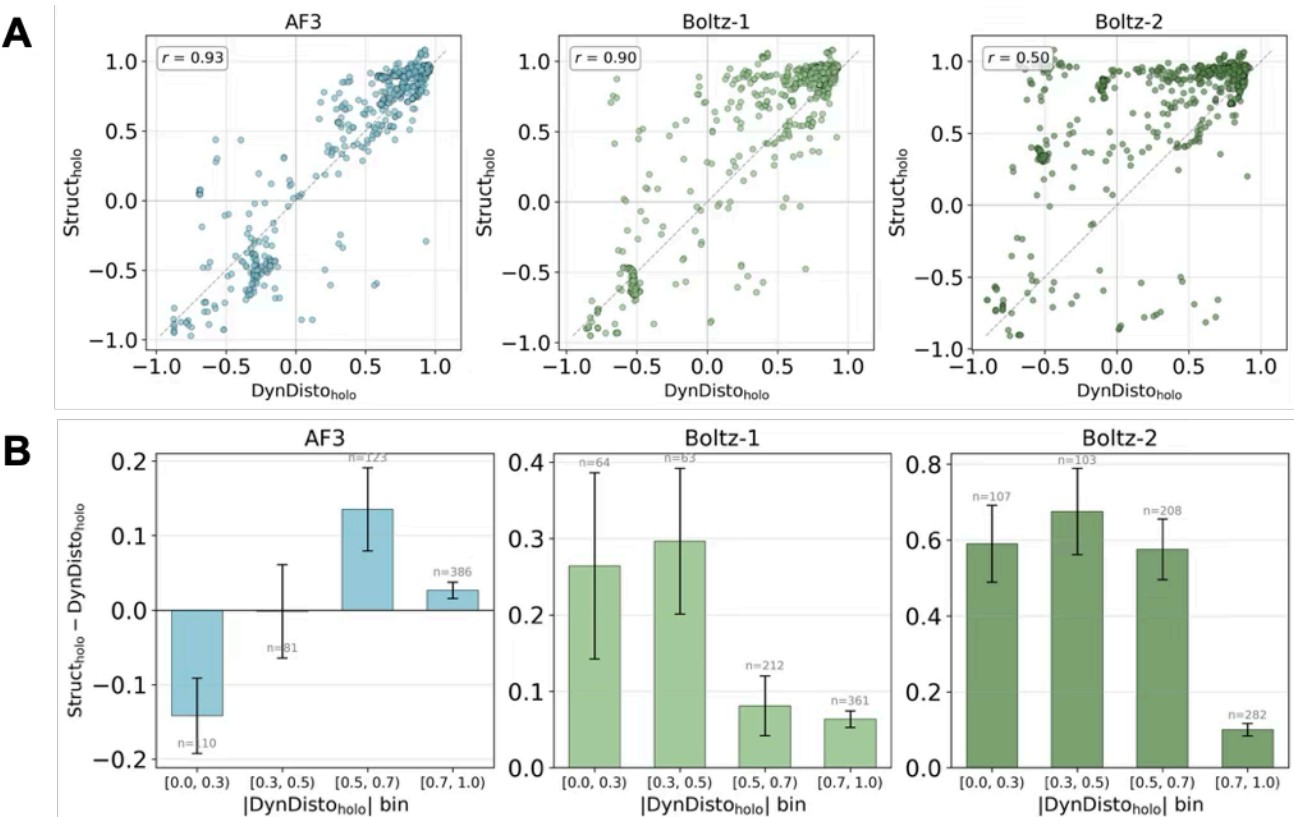

*Figure S6.* Quantitative analysis of systematic holo shift in structure predictions (ligand-induced set). **(A)** Scatter plots of $Struct_{holo}$ versus $DynDisto_{holo}$ for AF3, Boltz-1, and Boltz-2. Points above the diagonal ($y = x$) indicate that the structure module shifts predictions toward the holo state beyond what the distogram encodes. Pearson correlation coefficients are annotated per panel (AF3: $r = 0.93$; Boltz-1: $r = 0.90$; Boltz-2: $r = 0.50$). One-sided paired $t$-tests and Wilcoxon signed-rank tests for $Struct_{holo} - DynDisto_{holo} > 0$ confirm a significant positive shift for all three models (AF3: $t$-test $p = 3.24 \times 10^{-2}$, Wilcoxon $p = 1.32 \times 10^{-3}$; Boltz-1: $t$-test $p = 1.28 \times 10^{-24}$, Wilcoxon $p = 1.87 \times 10^{-33}$; Boltz-2: $t$-test $p = 7.30 \times 10^{-74}$, Wilcoxon $p = 1.08 \times 10^{-80}$), with 57.0%, 71.7%, and 86.0% of points above the diagonal, respectively. **(B)** Mean paired difference ($Struct_{holo} - DynDisto_{holo}$) binned by $|DynDisto_{holo}|$, with 95% confidence-interval error bars and per-bin sample sizes annotated. For Boltz-1 and Boltz-2, the holo shift is consistently positive across all bins and largest when distogram preference is weak ($|DynDisto_{holo}| < 0.7$), diminishing as $|DynDisto_{holo}|$ increases. For AF3, the structure module instead attenuates the holo signal when $|DynDisto_{holo}| < 0.3$ and contributes a positive shift only at intermediate distogram strengths ($|DynDisto_{holo}| \in [0.5, 0.7)$).

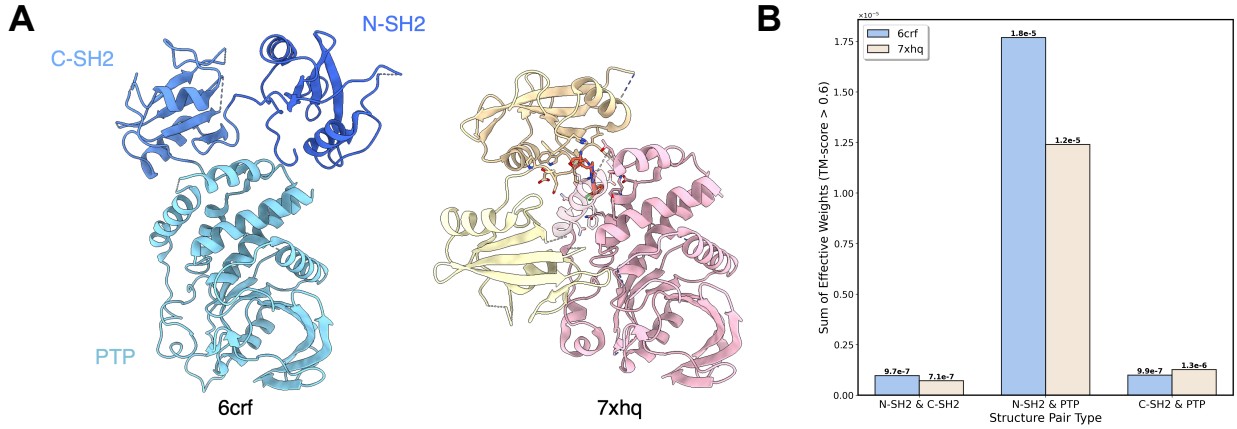

*Figure S7.* Case study results of the identified cluster. (A) Structures of the cluster highlighted by the black circle in Figure 3A, with domains annotated. The pair of PDB IDs 6crf and 7xhq was selected for analysis. (B) Number of structure-based search hits obtained using Foldseek, with queries constructed from selected two-domain permutations.

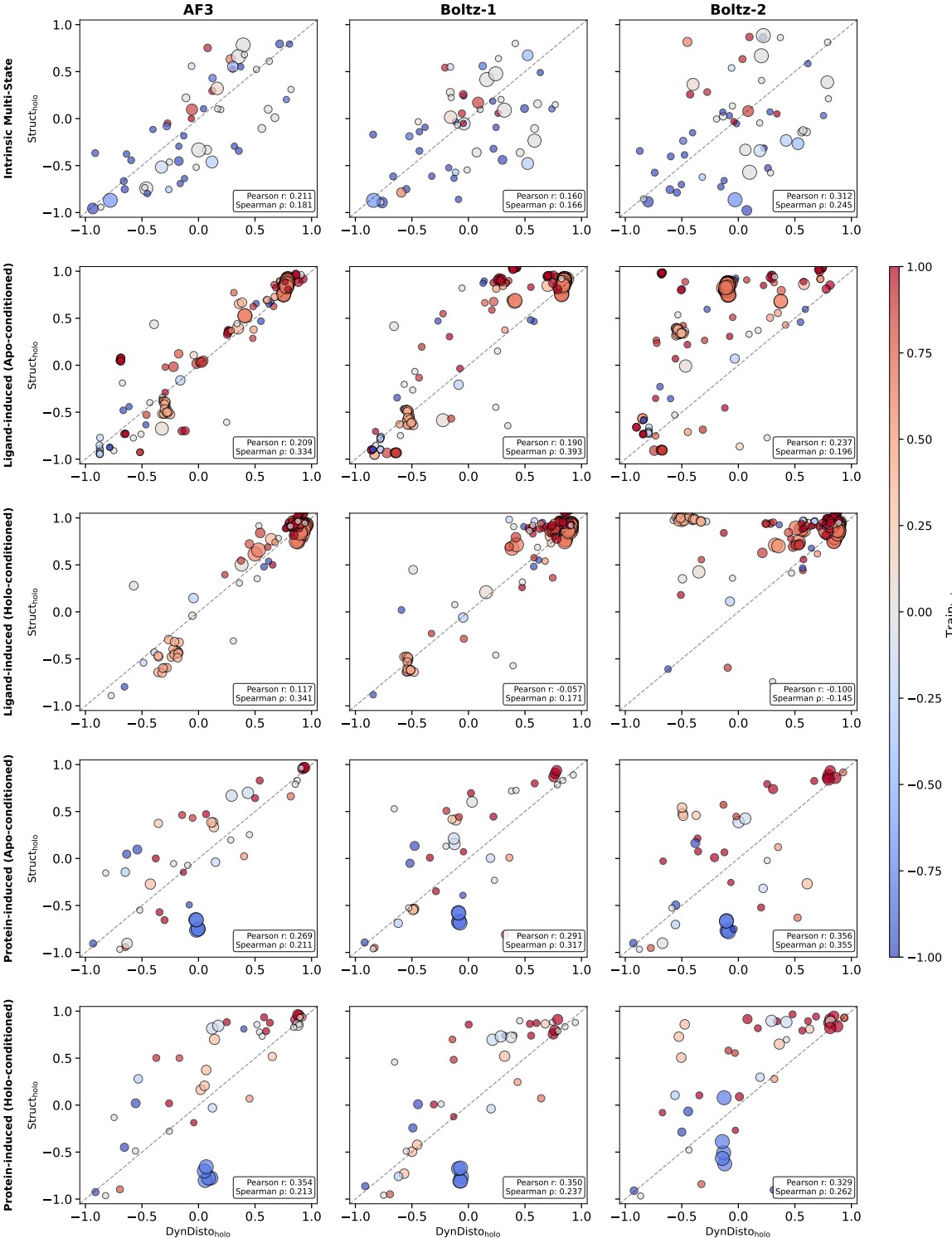

*Figure S8.* Model comparison of $DynDisto_{holo}$ vs. $Struct_{holo}$ across models (AF3, Boltz-2, Boltz-1), dataset categories(intrinsic multi-state, ligand-induced, protein-induced), and apo/holo-conditioned predictions. Points are colored by $Train_{holo}$ and sized by the number of searched entries in training data, with control constraint of $|MSA_{holo}| < 0.3$. Dashed line indicates $y = x$. Correlation coefficients (Pearson and Spearman) are computed between $Struct_{holo} - DynDisto_{holo}$ and $Train_{holo}$.

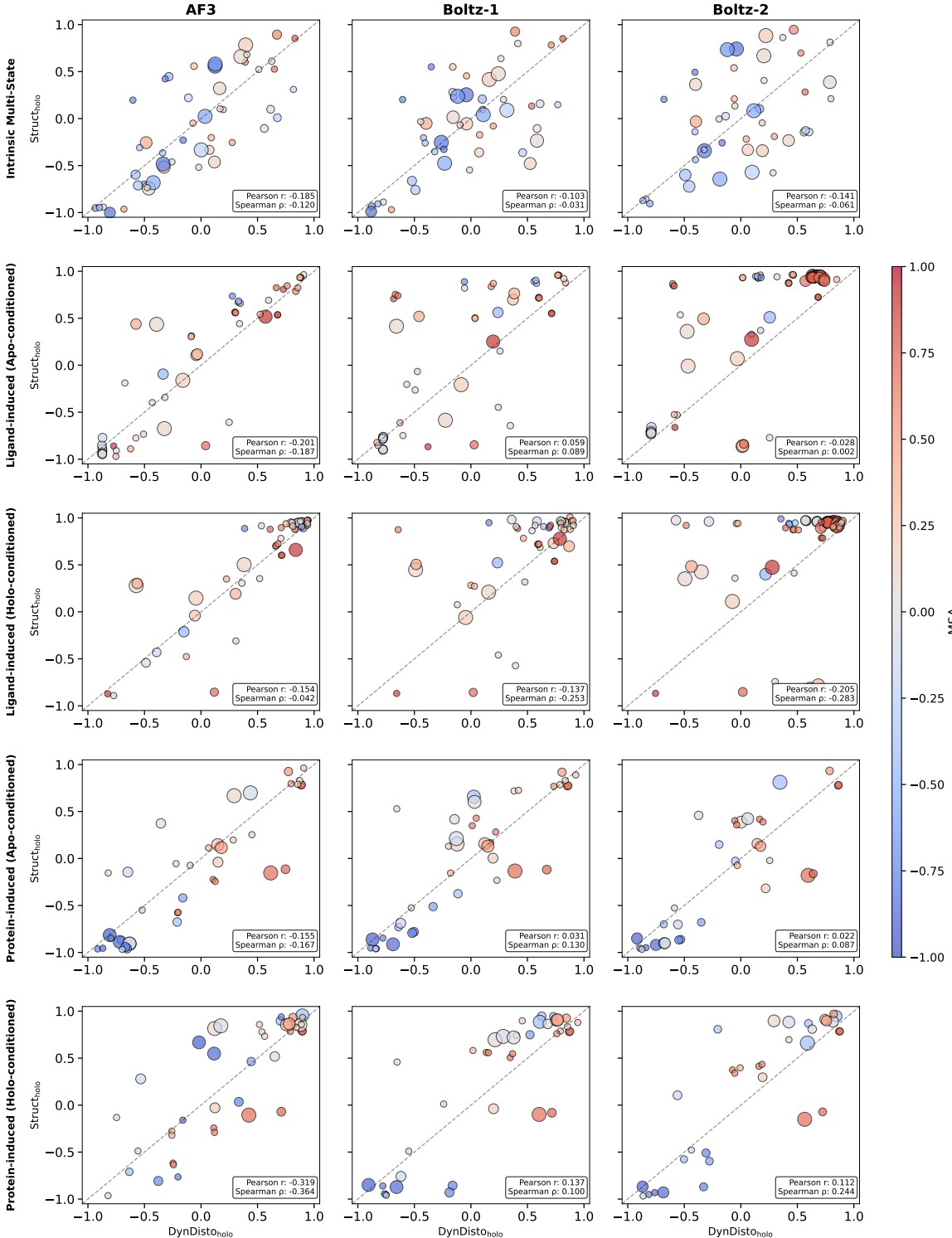

*Figure S9.* Model comparison of $DynDisto_{holo}$ vs. $Struct_{holo}$ across models (AF3, Boltz-2, Boltz-1), dataset categories(intrinsic multi-state, ligand-induced, protein-induced), and apo/holo-conditioned predictions. Points are colored by $MSA_{holo}$ and sized by the number of searched entries in training data, with control constraint of $|Train_{holo}| < 0.3$. Dashed line indicates $y = x$. Correlation coefficients (Pearson and Spearman) are computed between $Struct_{holo} - DynDisto_{holo}$ and $MSA_{holo}$.

