# OpenReview forum: "ProMiSE: Protein Multi-State Evaluation Benchmark in Biological Contexts"
_ICML.cc/2026/Conference — ICML 2026 regular_

### Official Review · Reviewer_hksL · 2026-02-22

**Soundness:** 3
**Presentation:** 3
**Significance:** 4
**Originality:** 2
**Overall Recommendation:** 3
**Confidence:** 4

**Summary:**

This paper introduces ProMiSE, the first unified benchmark for evaluating multi-state protein structure prediction within biologically relevant contexts, including intrinsic, ligand-induced, and protein-induced conformational changes. Through a comprehensive evaluation of state-of-the-art models like AlphaFold3 and Boltz-2, the authors identify a systematic "conformational collapse," where models often converge to a single dominant state—typically the holo-like conformation—even when conditioned on apo (partner-free) inputs. Their analysis reveals that this bias primarily emerges in the structure module rather than the pairwise representations, driven largely by the memorization of skewed training data. The study further validates this bottleneck by demonstrating that BioEmu, which fine-tunes the decoding module on diverse dynamics data, achieves significantly better conformational dispersion.

**Compliance With Llm Reviewing Policy:**

Affirmed.

**Key Questions For Authors:**

1. What is the intended definition of “dynamics” in this benchmark?
2. Can structure-module bias be mitigated without MD supervision?
3. Why is sampling success defined by existence rather than distributional coverage?

**Limitations:**

Yes.

**Strengths And Weaknesses:**

This paper definitely tackles one of the most important benchmarks in AI4biology: protein conformation landscapes. However, I am not sure if the "diversity" in this dataset is worth trusting if they are not validated from any computational simulation. I am not an expert in cluster-based analysis so I am willing to change my opinion if the author can persuade me.
### Strengths:
**Clear identification of the failure mode (structure module collapse)**:
The authors convincingly show that conformational collapse is introduced primarily by the structure decoding module, not by MSA signals or pair representations. The BioEmu comparison (same pairformer, different decoder training) is especially strong

**Well-curated, biologically contextual benchmark**:
ProMiSE is carefully constructed around biological mechanisms (intrinsic, ligand-induced, protein-induced), using native assemblies rather than artificial monomer decompositions

### Weakness:
**Unfair comparison:** BioEmu has an additional reinforcement learning stage which seems to introduce additional intrinsic dynamics but destory its capacity in generating native states. I am not sure if this accuracy tradeoff dynamics is inherent and would love to see some analysis over that.

**Dataset from actual simulations:** There are many publicly available MD simulations like DESRES, ATLAS, MDCath, and BioEmu. I would think its beneficial to include analysis on these dataset for real dynamical information.

---

> ### Author Rebuttal · Authors · 2026-03-31
>
> We thank the reviewer  for the constructive feedback. Detailed figures and tables referenced in this response are available at https://figshare.com/articles/conference_contribution/Promise_Benchmark_Rebuttal/31898266?file=63324760. We address each weakness (W1–W2) and question (Q1–Q3) below.
>
> Q1)
> We acknowledge that the term "intrinsic dynamics" set was imprecise, and we will revise it to **"intrinsic multi-state"** set  throughout the manuscript. Our intended focus is not on dynamics, but on whether models can recover functionally relevant conformational states—specifically, experimentally validated apo/holo structures that represent distinct functional contexts. The rationale for using such discrete PDB-based states as ground truth is that these structures correspond to (meta)stable free-energy minima, making them practically meaningful targets for evaluation. For a more detailed discussion on the relationship between discrete PDB-based evaluation and continuous conformational landscapes, see also our response to Reviewer 1, W4.
>
> Q2)
> We argue that structure-module bias can be mitigated even without MD supervision, as long as the **training data contains sufficient structural diversity.** We also note that MD data alone is not a complete solution, due to limitations in force field accuracy and sampling timescales.
> In BioEmu, pretraining on sequence-clustered AFDB explicitly teaches the diffusion model that one sequence can map to multiple conformations, achieving substantial coverage of functionally distinct states without any MD data (Lewis et al., Science, 2025, Figure 2). In contrast, AF3 and Boltz are trained on PDB data where most sequences appear with a single (often holo) structure, inducing conformational collapse. We therefore attribute structure-module bias primarily to the lack of balanced multi-state training examples, rather than the absence of MD supervision.
>
> Q3)
> We agree that distributional coverage is a more stringent and informative criterion. However, in many cases models fail to recover even a single structure corresponding to each known state, and accurately estimating the true distributional coverage of protein multi-state ensembles itself remains a challenging open problem.
> We therefore approximate distributional coverage using precision, defined as the hit rate—the fraction of samples achieving TM-score ≥ 0.8 to at least one ground-truth state—within clusters that satisfy our existence-based success criterion.
> As shown in Table R4, this hit rate is consistently moderate to high across models (~0.6–0.8), suggesting that once a model can access a state, it produces a substantial concentration of samples around that state. Overall, these results suggest that the **main bottleneck is discovering all relevant states rather than distributing samples within each state.**
> &nbsp;
> ### Table R4: Hit rate within successful clusters (partial distributional coverage).
> Values report the mean fraction of samples achieving TM-score ≥ 0.8 to at least one ground-truth state, with standard deviation across clusters. Best results are shown in bold.
>
> | Metric | AF3 | Boltz-1 | Boltz-2 | Chai-1 | BioEmu |
> |---|---|---|---|---|---|
> | Mean fraction (↑) | 0.818 | **0.840** | 0.617 | 0.804 | 0.767 |
> | Std (↓) | 0.320 | 0.302 | 0.371 | 0.290 | **0.234** |
> | # Clusters | 19 | 13 | 16 | 19 | 24
>
> W1)
> We agree that BioEmu's additional training stage(PPFT; property-­prediction fine-­tuning), may influence the balance between diversity and accuracy.
> In our analysis (Table R4), BioEmu achieves the highest number of state-covered clusters (24 vs. 13–19 for other models), confirming superior state coverage. However, within these clusters, BioEmu's mean hit rate (0.767) is lower than AF3 (0.818) or Boltz-1 (0.840), suggesting a **potential trade-off between covering multiple conformations and concentrating around native states**.
> We note that this trade-off cannot be directly attributed to PPFT, as intermediate checkpoints are not publicly available for controlled comparison.
>
> W2)
> We agree that MD-generated datasets provide useful complementary information on conformational landscapes.As a preliminary analysis, we compared model-generated ensembles against BioEmu testset (Lewis et al., Science, 2025) and observed that current structure prediction models tend to explore only a limited region of the conformational space, while BioEmu shows broader coverage (Figure R11).
> We observed that models trained primarily on static PDB structures (e.g., AF2/AF3-style training) do not produce meaningful structural diversity, whereas models explicitly trained on diverse conformations (e.g., BioEmu) show improved coverage.
> However, our primary focus is on recovering experimentally validated, functionally relevant states, which we believe provides a more direct and practical evaluation target.
>
> We will incorporate these revisions into the revised manuscript.

---

> > ### Author Rebuttal · Reviewer_hksL · 2026-04-03
> >
> > Thanks for the quick response. I have a few follow-up questions:
> >
> > **Q2**
> >
> > The question remains: generating diverse structures are easy with msa subsampling and lots of other computational efforts. However, it is important to know which ensemble conformations are valid and what are relative weights. Force field and MD has its issues but at least it provides a unified probablisitic description of boths. I still hardly believe that simply generating diverse structures are enough. Some validations are definitely needed.
> >
> > Otherwises this is an exiciting benchmark and I would definitely love to increase my score if authors can somehow provide more rebuttal on this.

---

> > > ### Author Response · Authors · 2026-04-05
> > >
> > > Additional figures and extended results can be found at the following link. https://figshare.com/articles/conference_contribution/Promise_benchmark_rebuttal_2/31941063?file=63464034
> > >
> > > We first thank the reviewer for additional feedback.
> > >
> > >  We agree that evaluating sampling existence alone is insufficient, and validating whether generated ensembles reflect physically meaningful conformational weights is also important.
> > >
> > > Since current generative models do not provide evaluation metrics that reflect the underlying conformational distribution, we instead devised two complementary analyses using PDB structures as references, as they represent energy minima. The objective was to assess 1) whether the structures with the highest relative weight are close to the reference structures, and 2) whether the structures ranked highest by the confidence metrics prioritize the reference structures.
> > >
> > > First, we measured how close the frequently sampled conformations are to PDB structures by determining the density percentile of each reference PDB structure on the PCA conformational landscape(Figure R1). This analysis showed that the conformations given the largest relative weight by the model did not sample the reference PDB structures(Figure R2), even though these structures would be expected to be sampled because they correspond to energy minima.
> > >
> > >   Across several models and benchmark subsets, confidence-based ranking does not consistently outperform random sampling, and in some cases performs comparably or worse(Figure R3). This indicates that current model confidence scores are not reliably calibrated to prioritize experimentally observed static conformations, even when such conformations are present among generated samples. The diminishing gap at larger k further suggests that sampling diversity plays a dominant role in overall success.
> > >
> > >   Therefore, we found that current generative models may occasionally sample experimentally observed valid structures, but they do not assign them high relative weight or high confidence within the conformational ensemble.

---

### Official Review · Reviewer_cYs4 · 2026-03-04

**Soundness:** 2
**Presentation:** 3
**Significance:** 3
**Originality:** 3
**Overall Recommendation:** 5
**Confidence:** 4

**Summary:**

This manuscript introduces a novel benchmark designed to evaluate the ability of SOTA protein structure prediction models to capture multi-state dynamics. The method integrates three distinct conformational change mechanisms—intrinsic dynamics, ligand-induced changes, and protein-induced changes—within a unified dataset derived from native biological assemblies. The authors evaluate several leading models, and highlight significant limitations in current models. Through detailed analysis, the study identifies the structure decoding module, influenced by training set memorization, as the primary bottleneck rather than the upstream pair representations.

**Compliance With Llm Reviewing Policy:**

Affirmed.

**Final Justification:**

I have no further issue.

**Key Questions For Authors:**

1. The current evaluation is based on 100 samples per entry. In actual drug screening, the hit rate of the model in generating effective conformations is key. It would be beneficial if the authors discuss the relationship between sampling quantity and success rate.
2. Although TM-score is high, they need evaluate whether the generated structures have local stereochemical conflicts (Steric Clashes) . It is recommended that future versions integrate physical validity checks.
3. For large protein assemblies, performing all-to-all TM-score calculations and clustering(1,333,459 initial pairs) requires significant computational resources. While this is only the cost during dataset construction, it may pose a barrier for the community to independently update datasets.

**Limitations:**

No. The authors need discuss the limits.

**Strengths And Weaknesses:**

Strength
1.  The study creats a unified dataset covering intrinsic, ligand-induced, and protein-induced transitions that addresses a gap in the field. The rigorous curation process, including filtering for crystal artifacts and ensuring biological relevance, adds significant value.
2. The paper introduced novel bias quantification scores, which allows the authors to pinpoint where in the model pipeline the failure occurs (i.e., the structure module vs. pair representation), providing actionable insights for future model development.
3. The comparison with BioEmu effectively validates the hypothesis that fine-tuning the structure module on diverse data can alleviate conformational collapse, offering a clear path forward for improving dynamics-aware modeling.

Weaknesses
1. The current evaluation is based on 100 samples per entry. In actual drug screening, the hit rate of the model in generating effective conformations is key. It would be beneficial if the authors discuss the relationship between sampling quantity and success rate.
2. Although TM-score is high, they need evaluate whether the generated structures have local stereochemical conflicts (Steric Clashes) . It is recommended that future versions integrate physical validity checks.
3. For large protein assemblies, performing all-to-all TM-score calculations and clustering(1,333,459 initial pairs) requires significant computational resources. While this is only the cost during dataset construction, it may pose a barrier for the community to independently update datasets.

---

> ### Author Rebuttal · Authors · 2026-03-31
>
> We thank the reviewer  for the constructive feedback. Detailed figures and tables referenced in this response are available at https://figshare.com/articles/conference_contribution/Promise_Benchmark_Rebuttal/31898266?file=63324760. We address each weakness (W1–W3) below.
>
> W1)
> We agree that the relationship between sampling budget and success rate is important. In our experiments, we use a total budget of 10 seeds × 10 samples. To better understand this trade-off, we separately analyze (i) increasing the number of samples while fixing the number of seeds (10), and (ii) increasing the number of seeds while fixing the number of samples (10) (Table R1). For BioEmu, varying random seeds is not supported, so we only report results with a single seed while increasing the number of sampled structures. We observe that success rates consistently improve with increased sampling, but exhibit diminishing returns beyond a moderate number of samples (e.g., ~5–10), suggesting that moderate sampling is sufficient to capture most practical gains.
>
> W2)
> We **evaluated physical validity using MolProbity** (Williams et al., 2018), which provides MolProbity score integrating clashscore, Ramachandran outliers, and rotamer outliers (Table R2-3). Our analysis shows that AF3 and Boltz2 consistently achieve strong stereochemical quality across all tested categories, with AF3 in particular exhibiting low clashscores. These results confirm that the conformational sampling limitations are not driven by physically invalid structures. We will include these results in the supplementary information of the revised manuscript.
>
> W3)
> The **computational cost is modest in practice**, since clustering is performed only within typically small sequence clusters. In practice, the full pipeline is manageable (e.g., ~9 hours on 128 CPUs), and we provide the code to facilitate reproducibility and future updates.
> &nbsp;
> We will incorporate these revisions into the revised manuscript.
>
> [References]
> Williams, Christopher J., et al. "MolProbity: more and better reference data for improved all‐atom structure validation." Protein science 27.1 (2018): 293-315.

---

> > ### Author Rebuttal · Reviewer_cYs4 · 2026-04-05
> >
> > I've increased the score.

---

> > > ### Author Response · Authors · 2026-04-05
> > >
> > > Thank you for the update and we appreciate your positive assessment.

---

### Official Review · Reviewer_M4ZX · 2026-03-13

**Soundness:** 3
**Presentation:** 3
**Significance:** 4
**Originality:** 3
**Overall Recommendation:** 5
**Confidence:** 4

**Summary:**

This paper proposes a new multi-state benchmark of sequence-to-structure prediction, ProMiSE. Through careful curation and filtering of sequence-structure data available in Protein Data Bank (PDB), the authors were able to distill clusters of highly similar sequences with lowly similar deposited structures to serve as a basis of this benchmark dataset generation. Through an intra-cluster combination of these, the authors prepared pairs of (sequence, structure) pairs to serve as the data unit of the benchmark, i.e. consisting of an apo (sequence, structure) pair and a holo pair. Altogether, the dataset includes such pairs in three biological contexts that commonly yields conformational changes: intrinsic dynamics, ligand-induced changes, protein-induced changes. Success in this benchmark is measured by the structure prediction model’s ability to predict an apo structure given the apo sequence, and vice versa for holo, but not mixing both. State of the art structure prediction models (AF3, Boltz-1, Boltz-2, Chai-1, BioEmu with monomer limitation) have been benchmarked on this dataset. Furthermore, in order to understand the modes of failure, the authors have proposed an extensive set of conformational bias scores at the levels of structure prediction model input (MSA, Train) and output (DynDisto, Struct). In the experiments related to bias determination, the paper evidenced that (1) the structure module in these predictor models contribute to a bias towards holo structures (i.e., large Struct - DynDisto score), even if such bias only exists to a limited extent in the distograms of the models, (2) once controlled by one input factor of bias (MSA) to show the influence of another factor (Train), this structural bias is attributed to the skewed distribution of training data towards holo-states. Finally, the spread of Struct score at different DynDisto levels, across different random seed/samplings of the structure prediction, is developed as a measure of how (un)biased the structure module is. Comparisons with BioEmu, which has a higher diversity of structural training data through MD simulation fine-tuning, highlight that such diversity in training helps to increase the spread of the Struct score, hence reducing the bias.

**Compliance With Llm Reviewing Policy:**

Affirmed.

**Final Justification:**

The rebuttal has addressed my concerns, and reinforced my accept recommendation. I have accordingly increased my significance score from 3 to 4. As the structure prediction models are growing in maturity, benchmarks with multiple conformational states yield themselves as important next scientific goalposts. This paper and the related benchmark present the shortcomings of the SOTA models with creative metrics and analyses. Following the rebuttal revisions, these metrics and analyses are now strengthened in their statistical significance. They could be highly relevant for future structure prediction model development.

**Key Questions For Authors:**

(Q1) Are there any PDB entries that the benchmark dataset contains, hence reflected in Train scores, but protein structure prediction model training procedures do not include? If per-entry analysis is not feasible, what is a precautionary protocol that the authors propose to assert that whatever benchmark entry is taken into account in the Train score was actually used in the training of the model?

(Q2) In bias analysis (Sec 3.3), what is the reasoning behind picking 0.3 and 0.7 as metric thresholds? They seem somewhat arbitrary, if they are indeed arbitrary, can a sweep be made between [0.2, 0.4] to reduce the dependency of the claims made to this threshold?

(Q3) In Figure 3D, can the vertical spreads per model be quantified?

(Q4) In Figure S4 [also 3A], there seems to be a consistent island around DynDisto of -0.1 and Struct of 0.9 common across models. What is this island, do the pairs there share a functional similarity?

**Limitations:**

I follow the discussion of the present capabilities of the structure prediction models. However, I would have appreciated seeing limitations of the benchmark dataset itself. Such discussion could be about: categorization of the biological contexts of conformational diversity, categorization of structure entries with respect to the determination technique used and its resolution, choices made and their impacts in dataset filtering steps, shortcomings of the evaluation and bias inspection metrics proposed, and so on.

**Strengths And Weaknesses:**

Strengths

(S1) As the protein structure models become more successful and start to saturate the static structure prediction tasks, the next step of multi-state prediction and evaluation is growing in importance at great pace. This benchmark dataset, combined with tools of evaluating failures in conformational state prediction mode collapse, has the potential to immediately influence the field positively by providing both (1) a reference for practitioners who face conformational heterogeneity in their research and (2) a guideline for structure predictor developers to take conformational state diversity into account in next versions of structural models.

(S2) The filtering steps of the dataset has been explained in depth (especially with the algorithm tables in the Appendix A), the state of the art methods have been benchmarked, and their failures are investigated in depth with conformational bias analysis metrics proposed.

(S3) Correlation of the bias in structure modules of the prediction models with the bias of the training data is a meaningful scientific insight, and well-supported by the experiments reported.

Weaknesses

(W1) The Train score needs careful detailing. The benchmark dataset was created by a new and contemporary filtering of the PDB, and the authors mentioned that the per-model training cutoff date is taken into account (Sec. 2.4.4). However, it is not clear to me that even if an entry was deposited before the cutoff date, it is guaranteed to be in the training dataset of these models, since their training procedures also commonly feature custom curations from the PDB.

(W2) This is about the analysis of the model failure in the protein-induced subset (Sec. 3.2). It was my understanding that in the dataset curation process, for a given sequence cluster, the structures far away from each other (i.e., TM-score < 0.8) were chosen to form the pairs. Therefore, it could be the case that there exists structures, belonging to the same cluster, which are in fact intermediary states between the two extremal states chosen to construct a pair. Therefore, the observation that the predictors generate an intermediate conformation between the apo/holo pair could simply be an artifact of dataset generation, and the expected behavior for the predictor could indeed be to produce that intermediate conformation due to abundance of training data in that state.

(W3) Across the results presented (Figures 3A, 3D, S2B, S2C, S4, S5, S6), there are claims of correlation and spread supported only by the reader’s visual analysis by eye. In fact, by looking at some of the examples (S2B, S2C, S4-AF3), I cannot in full confidence claim the existence of such correlation. In a similar spirit, in 3D, I cannot see that BioEmu has a noticeably wider spread of Struct score, compared to Boltz models. Quantitative evidence (e.g., correlation coefficients) could have been quite helpful.

---

> ### Author Rebuttal · Authors · 2026-03-31
>
> We thank the reviewer for the constructive feedback. Detailed figures and tables referenced in this response are available at https://figshare.com/articles/conference_contribution/Promise_Benchmark_Rebuttal/31898266?file=63324760. We address each weakness (W1–W3) and question (Q1–Q4) below.
>
> W1 & Q1)
> We have revised Train_holo by accounting for **model-specific curation filters** and **weighted sampling** applied during training to address this concern.
> We agree that PDB deposition before a model’s training cutoff does not guarantee actual inclusion in its training set. Training pipelines often involve additional curation and weighted sampling; raw PDB entry counts may not accurately reflect the effective training exposure of individual entries.
> Our revised Train_holo addresses this by: (1) restricting the search space to entries passing each model's curation criteria (2) retaining only entries with high sequence identity (>0.8) and structural similarity (TM-score >0.9) to ensure conformational-state matching; and (3) computing "effective hits" by summing the sampling weights under the AF3/Boltz reweighting scheme(Abramson et al., 2024; Passaro et al., 2025).
> Under this formulation, Train_holo represents an estimate of how much training exposure existed for a given entry under each model's training regime. Importantly, our conclusions remain unchanged under the revised metric (Figure R1, R8): training set skew drives the structure module to produce biased structures.
>
> W2)
> We agree that restricting evaluation to apo/holo pairs with TM-score < 0.8 may overlook the possibility that intermediate conformations can also represent valid apo states. However, our analysis suggests that such cases have negligible impact on the overall evaluation.
> Since the multiple apo structures can exist within the same sequence cluster, we revisited failure cases in the protein-induced subset where apo structure was classified as failed. We re-evaluated predictions against all apo structures prior to TM-score-based clustering (< 0.8). Under this expanded evaluation, a small fraction of cases were rescued (AF3: 2/16, Boltz-1: 1/19, Boltz-2: 2/19, Chai: 1/14).
> This indicates that **most failures cannot be explained by alternative apo conformations present in the PDB**. Rather than representing meaningful intermediates, these predictions are better characterized as incorrect structures that do not match any known apo state in the PDB. We will revise the manuscript accordingly.
>
> W3)
> We have added formal statistical tests and quantitative metrics for all figures where claims previously relied on visual inspection. Detailed statistics are provided in each figure caption; here we summarize the key findings.
> For Figure 3D, we quantified the vertical spread as the within-distogram standard deviation of Struct_holo. BioEmu produces significantly more diverse structures per distogram than all other models (Wilcoxon signed-rank, p < 0.0001; Figure R2-A,B) and consistently outperforms others in spanning both conformational states (Figure R2-C,D).
> For Figures S2B and S2C, we added Pearson and Spearman correlation coefficients, confirming moderate correlations between monomer modeling quality and complex prediction quality. Also binned success rates further confirm a clear monotonic decline as monomer modeling quality decreases. (Figure R3)
>
> For Figure S4, paired statistical tests confirm that the structure module systematically shifts predictions toward the holo state beyond what the distogram encodes (p < 10⁻¹⁰ for all models; Figure R4).
> For Figures S5 and S6, training data exposure (Train_holo) is positively correlated with structural bias (Struct_holo) across all subsets and models. We additionally provide corresponding plots using the revised Train_holo, which also show consistent trends (Figure R7-10).
>
> Q2)
> To address this, we examined how the paired difference (Struct_holo − DynDisto_holo) varies across |DynDisto_holo| bins (Figure R4B), and observed a sharp decrease around the 0.7 threshold, empirically supporting it as a meaningful boundary rather than an arbitrary choice.
> The thresholds were chosen to balance robustness and sample size. We performed a sensitivity sweep in [0.2, 0.4], and observed consistent trends of correlation between Train_holo and Struct_holo (Figure R5-6).
>
> Q3)
> As detailed in W3, we quantify the vertical spread using within-distogram standard deviation. This confirms that BioEmu produces significantly greater structural diversity (Figure R3).
>
> Q4)
> We observed that this island is primarily composed of T4 hydrolase (23 pairs), all sharing the same apo-structure (PDB ID: 7dws), along with a few sugar-binding proteins and calmodulins (e.g., PDB IDs: 2uvi, 4ieb). Despite binding various ligands, their holo-structures remain within the same conformational cluster, forming the island.
> &nbsp;
> We will incorporate these revisions into the revised manuscript.

---

> > ### Author Rebuttal · Reviewer_M4ZX · 2026-04-04
> >
> > I would like to thank the authors for their detailed rebuttal, and I have adjusted my scores accordingly. The additional statistical analyses, figures, and the Train_holo adjustments strengthen the paper’s claims. I strongly recommend adding a discussion of the benchmark dataset with its limitations and future directions at the end or in the appendices (refer to my review’s “Limitations” section above).

---

> > > ### Author Response · Authors · 2026-04-05
> > >
> > > We thank the reviewer for this constructive suggestion and agree that the limitations and possible future work should be stated. We plan to report the limitations and possible future works on the revised manuscript.
> > >
> > > First, the biological contexts of conformational change are often overlapping, and some entries do not fit cleanly into a single category as in our current setup. Although it requires careful curation to isolate protein-induced and ligand induced changes, we plan to include multimer assemblies with ligand contexts in our future work.
> > >
> > > Second, the current ligand-induced set includes only monomeric assemblies and therefore does not capture mixed-context cases, such as multimeric ligand-induced transitions or systems of simultaneous ligand and protein binding. Expanding the benchmark to include such mixed-context systems is an important direction for future work.
> > >
> > > Third, our current pair-extraction criterion (TM-score < 0.8) is likely too stringent and may preferentially retain only large conformational changes. As a result, more localized but still biologically important motions, such as GPCR TM6 movement or transporter pocket motions, are mostly dropped during the curation process. In future versions of the dataset, we plan to revise this criterion to better capture these functionally relevant conformational changes.
> > >
> > > Finally, the current evaluation setup also has limitations. In particular, the pairwise evaluation does not account for other entries within the same cluster that may also represent valid ground-truth structures for the same sequence. One possible future direction is to evaluate distributional coverage and modify the bias evaluation accordingly. This is beyond the scope of the current revision, but we agree that it would be a valuable direction for future work.

---

### Official Review · Reviewer_7vUF · 2026-03-15

**Soundness:** 3
**Presentation:** 3
**Significance:** 3
**Originality:** 3
**Overall Recommendation:** 4
**Confidence:** 4

**Summary:**

This work introduces ProMiSE, a benchmark designed to evaluate multistate protein structure prediction across three major conformational regimes: intrinsic dynamics, ligand-induced changes, and protein-induced conformational transitions. The transition from predicting static protein structures to generating dynamic structural ensembles is of importance in the post-AlphaFold era. The evaluation reveals systematic limitations in current models. This is a timely and interesting piece of work.

**Compliance With Llm Reviewing Policy:**

Affirmed.

**Key Questions For Authors:**

See above.

**Limitations:**

yes

**Strengths And Weaknesses:**

**Strengeths**

1.ProMiSE is a comprehensive benchmark for systematically evaluating multi-state protein structure prediction, it's timely.

2.The work provides a systematic evaluation of current methods, identifies several limitations in existing models, and offers insights into potential directions for improvement.

3.The manuscript is well written and easy to follow.

**Weakness**
1. The evaluation of dynamic properties in this paper relies heavily on aligning predictions with discrete apo and holo states. However, successfully generating a specific apo or holo structure does not necessarily equate to capturing the true conformational ensemble or the underlying energy landscape. Could the authors provide further evidence or justification on the reliability and feasibility of evaluating generative models using this dual-state setup? How well does this metric reflect a model's true capability in ensemble generation?

2. The dataset categorizes structural changes into three distinct subsets: *Intrinsic Dynamics*, *Ligand-Induced*, and *Protein-Induced*. Is this strict categorization robust enough to capture the complexity of real biological systems? Specifically, does this setup account for allosteric regulation, where a ligand binding at one site induces conformational changes at a distant site (often a mixture of ligand-induced and intrinsic mechanisms)? It would be helpful if the authors could clarify whether allosteric data is included and how it is classified.

3. In Section 3.4, the authors utilize BioEmu to demonstrate that adjusting the structure module and training data can enable generative models to sample diverse structures. Is it entirely reasonable to generalize this conclusion to the other models tested? BioEmu is built upon an AlphaFold2-like representation base, whereas the other state-of-the-art models evaluated (e.g., AF3, Boltz-1/2, Chai-1) utilize fundamentally different diffusion-based architectures. Can the insights derived from an AF2-based model directly explain the "conformational collapse" observed in modern diffusion models?

4. The paper wants to evaluate the models' ability to generate structural ensembles and capture "Intrinsic Dynamics." However, relying solely on discrete PDB structures (apo/holo snapshots) as the ground truth seems insufficient. PDB structures are isolated, static snapshots that do not represent the continuous conformational energy landscape (Boltzmann distribution) of a protein in solution.  Without comparing the generated structural diversity against Molecular Dynamics (MD) trajectories or free energy surfaces, it is difficult to determine whether the generated "diversity" represents true physical transitions or merely unphysical noise. Furthermore, PDB pairs cannot evaluate whether the models correctly capture the relative populations (statistical weights) of different states. Could the authors explain why MD simulations or NMR ensembles were not included as continuous benchmarks, and how the current discrete metric justifies the claim of modeling true "dynamics"?

I would be happy to raise my score if the authors can address some of the issues above.

---

> ### Author Rebuttal · Authors · 2026-03-31
>
> We thank the reviewer  for the constructive feedback. Detailed figures and tables referenced in this response are available at https://figshare.com/articles/conference_contribution/Promise_Benchmark_Rebuttal/31898266?file=63324760. We address each weakness (W1–W4) below.
>
> W1)
> We agree that our current setup does not capture the full conformational ensemble or energy landscape. Our objective, however, is not to model the full ensemble, but to assess whether models can recover **functionally relevant conformational states associated with physiological context**. We provide a more detailed discussion on the rationale for using PDB-based ground truth in our response to W4.
>
>
> W2)
> We agree that real biological systems often involve mixed mechanisms such as allostery. Our categorization is intentionally designed to isolate dominant biological contexts (intrinsic, ligand-, or protein-induced) to enable controlled evaluation. Allosteric cases triggered by ligand binding are **included within the ligand-induced subset**, as we classify based on the external context rather than the underlying mechanistic pathway. For example, the 6z66 (apo) / 6zin (holo) pair, which is included in the ligand-induced set, represents an allosteric GPCR system where agonist binding at one site induces conformational changes at a distant location. We acknowledge that more complex mixed-context systems are important and represent a direction for future extensions.
>
>
> W3)
> We clarify that BioEmu uses a **diffusion-based** structure decoder similar to AF3 and Boltz.
> Also, the representation modules in BioEmu (AF2-based frozen Evoformer) and other AF3 based architectures (Pairformer) share similar architectures that extract structural constraints from MSA-derived coevolutionary signals.
> Since the only meaningful difference is that BioEmu trains its diffusion module on conformationally diverse data, we can generalize modifying the structure decoder’s training data distribution is a key to restore conformational diversity.  We will revise Section 3.4 to clarify this point.
>
> W4)
> We agree that discrete PDB structures do not represent the full conformational ensemble and underlying energy landscape. Although capturing the conformational ensemble is important, true dynamical trajectories and free-energy surfaces are generally unavailable. NMR ensembles exist for only a limited range of proteins, while MD-based references are constrained by force-field accuracy and sampling timescales.
> Our objective is therefore not to model or evaluate the full ensemble, but to **focus on experimentally validated conformational states with functional relevance**. Accordingly, we use PDB structures as ground truth for these states. We note that experimentally determined crystal and cryo-EM structures are not arbitrary snapshots—they represent (meta)stable states occupying local or global free-energy minima, where different experimental conditions can trap proteins in distinct energy wells (Henzler-Wildman et al., 2007; Ellaway et al., 2024). For this purpose, curated PDB pairs provide a well-grounded and practically meaningful reference.
> While full ensemble evaluation is beyond the scope of our current work, we conducted a preliminary evaluation based on MD dataset to examine whether current models cover MD-generated landscape. (see Figure R11 and our response to Reviewer 4, W2).
> We acknowledge that the term "intrinsic dynamics" set was imprecise in this context, and we will revise it to "intrinsic multi-state" set to avoid implying full dynamical ensemble-level characterization.
> &nbsp;
> We will incorporate these revisions into the revised manuscript.
>
> [References]
> 1. Henzler-Wildman, Katherine, and Dorothee Kern. "Dynamic personalities of proteins." Nature 450.7172 (2007): 964-972.
> 2. Ellaway, Joseph IJ, et al. "Identifying protein conformational states in the Protein Data Bank: toward unlocking the potential of integrative dynamics studies." Structural Dynamics 11.3 (2024).

---

> > ### Author Rebuttal · Reviewer_7vUF · 2026-04-05
> >
> > Thank you for the feedback and the adjustments. I do not have any additional concerns or questions.

---

### Decision · Program_Chairs · 2026-04-30

**Decision:**

Accept (regular)

**Comment:**

This paper presents an important direction for evaluating protein structure ensemble prediction methods. It is a valuable resource and all reviewers unanimously vote for acceptance of this paper.